# Highlighting What Matters: Promptable Embeddings for Attribute-Focused Image Retrieval

**Siting Li**
University of Washington
sitingli@cs.washington.edu

**Xiang Gao**
IIIS, Tsinghua University
x-gao22@mails.tsinghua.edu.cn

**Simon Shaolei Du**
University of Washington
ssdu@cs.washington.edu

## Abstract

While an image is worth more than a thousand words, only a few provide crucial information for a given task and thus should be focused on. In light of this, ideal text-to-image (T2I) retrievers should prioritize specific visual attributes relevant to queries. To evaluate current retrievers on handling attribute-focused queries, we build COCO-FACET, a COCO-based benchmark with 9,112 queries about diverse attributes of interest. We find that CLIP-like retrievers, which are widely adopted due to their efficiency and zero-shot ability, have poor and imbalanced performance, possibly because their image embeddings focus on global semantics and subjects while leaving out other details. Notably, we reveal that even recent Multimodal Large Language Model (MLLM)-based, stronger retrievers with a larger output dimension struggle with this limitation. Hence, we hypothesize that retrieving with *general* image embeddings is suboptimal for performing such queries. As a solution, we propose to use *promptable* image embeddings enabled by these multimodal retrievers, which boost performance by highlighting required attributes. Our pipeline for deriving such embeddings generalizes across query types, image pools, and base retriever architectures. To enhance real-world applicability, we offer two acceleration strategies: Pre-processing promptable embeddings and using linear approximations. We show that the former yields a 15% improvement in Recall@5 when prompts are predefined, while the latter achieves an 8% improvement when prompts are only available during inference.

## 1 Introduction

Images offer valuable information that facilitates problem-solving and reasoning [Zellers et al., 2019, Lu et al., 2022, 2023]. Although there may be abundant information in one image, especially when it depicts a complex scene with plenty of elements [Gabbay et al., 2021, Urbanek et al., 2024, Nguyen et al., 2024], usually only a small part is critical to a task or query at a time. For both performance and efficiency considerations, there has been recent effort to focus the models on key aspects of images for better visual reasoning [Wu and Xie, 2023, Hu et al., 2024b, OpenAI, 2025]. Similarly, for text-to-image (T2I) retrieval that helps knowledge and fact checking [Yasunaga et al., 2022, Sharifymoghaddam et al., 2024], an ideal retriever should also be able to select images with given attributes of interest, such as a specified time, location, or object, which are not necessarily the main elements of the image.

---

 https://github.com/lst627/COCO-Facet

39th Conference on Neural Information Processing Systems (NeurIPS 2025).

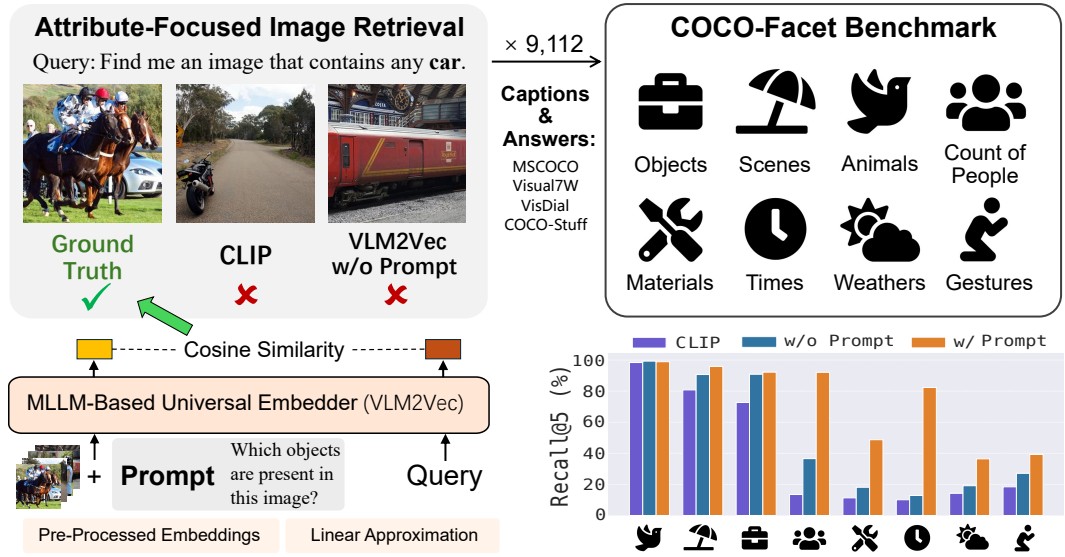

Figure 1: **Overview.** *(Above)* We study the task of attribute-focused text-to-image retrieval and build COCO-FACET for benchmarking various retrievers. *(Below)* We show that using promptable image embeddings enhances performance on such queries, and propose two acceleration strategies to improve its applicability.

*Are current T2I retrievers capable of accomplishing attribute-focused queries?* We explore this question on various CLIP-like models, from CLIP [Radford et al., 2021] to recent SigLIP2 [Tschannen et al., 2025], as well as Multimodal Large Language Model (MLLM)-based embedders like VLM2Vec [Jiang et al., 2024b]. Since commonly used T2I benchmarks like MSCOCO [Lin et al., 2014] and Flickr30K [Young et al., 2014] only contain general queries on global alignment, we build a new benchmark, COCO-FACET with 9,112 attribute-focused queries of 8 types, based on existing annotations of COCO images [Lin et al., 2014, Zhu et al., 2016, Caesar et al., 2018, Das et al., 2017]. We find that current retrievers behave worse on attributes other than `Animals` compared with general T2I retrieval on MSCOCO, and struggle more with detailed or underexplored attributes such as `Time`. Additionally, we discover that they fail to prioritize images with the correct-but-non-dominant attribute over images with a wrong-but-dominant attribute (See an example in Figure 1).

As embedders with various architectures and scales all fall short on such queries, we hypothesize that retrieving with *general* image embeddings is inefficient and suboptimal for this task. Therefore, we propose to retrieve with *promptable* image embeddings: We employ MLLM-based universal embedders that can process combination of images and text, and show that using the GPT-written prompt for each category as text helps to highlight key attributes in image embeddings, demonstrating improvement on harder attributes while maintaining good performance on easier ones (See the bar plot in Figure 1). Additionally, we design two strategies to accelerate this pipeline: Predefining potentially useful prompts and pre-processing the promptable image embeddings, or deriving linear approximation of the embedder at test time, which can be efficiently applied to the query vectors.

Our main contributions are listed as follows:

- Introduced in Section 3, our benchmark COCO-FACET on attribute-focused T2I queries is a good supplement to the current general-purposed T2I retrieval evaluation. We also provide construction pipelines, which can be utilized for building benchmarks focusing on other attributes for future research.
- In Section 3.2, we reveal the limitation of current CLIP-like retrievers and MLLM-based embedders on attribute-focused T2I queries, which affects models with various scales, image resolutions, and output dimensions.
- In Section 4, we propose to use promptable image embedding enabled by MLLM-based universal embedders as the solution. We show that it enhances the retrieval performance and generalizes over query types, image pools, and base retriever architectures.
- In Section 5, we develop two acceleration strategies for real-world usage. Our results demonstrate that the pre-processing technique increases Recall@5 by 15% when prompts are predefined, while

the linear approximation achieves an 8% improvement on Recall@5 when prompts are only available at inference time.

## 2 Related Work

**Text-to-image Retrieval** has been a long-standing research direction not only due to the real-world need for image search, but also as an important step in general problem-solving [Yasunaga et al., 2022, Hu et al., 2024a]. CLIP-like dual-encoder approaches [Radford et al., 2021, Zhai et al., 2023, Cherti et al., 2023, Li et al., 2023b] are widely-used due to their efficiency and remarkable zero-shot performance on standard T2I benchmarks like MSCOCO [Lin et al., 2014] and Flickr30K [Young et al., 2014]. Recent MLLM-based retrievers [Lin et al., 2024, Lan et al., 2025, Zhou et al., 2024a, Jiang et al., 2024b] extend the input modality to joint image-text pairs. Since these models output image and text embeddings for retrieval, there have been benchmarks for more comprehensive evaluation of embeddings [Wei et al., 2024, Jiang et al., 2024b, Xiao et al., 2025], which include domain-focused retrieval subsets like FashionIQ [Wu et al., 2021], EDIS [Liu et al., 2023], OVEN [Hu et al., 2023], Wiki-SS-NQ [Ma et al., 2024]. We note that these subsets either focus on the global semantics of the images or pre-select images with a single main subject, though in-the-wild images can be visually crowded with many attributes. Visual Genome [Krishna et al., 2017] considers such complexity by annotating the images with detailed descriptions for advanced visual understanding. COCO-Attributes [Patterson and Hays, 2016] is built with attribute annotations for multi-label classification but only targets main subjects like people, animals, and objects.

On the other hand, previous studies on visual grounding and reasoning have pointed out that CLIP-like models [Paiss et al., 2023, Chen et al., 2024a, Tong et al., 2024] might neglect visual details. Sogi et al. [2024] shows that the retrieval performance degrades when the target objects are small. To overcome this limitation, researchers propose to use two-stage approaches [Miech et al., 2021, Geigle et al., 2022] or guidance from an LLM [Lee et al., 2024]. Notably, pre-CLIP methods represented by scene graphs [Johnson et al., 2015, Schuster et al., 2015, Pham et al., 2024] might capture more visual details, but they lag behind in terms of data availability and inference efficiency.

Another improvement based on CLIP-like models is to obtain task-specific image embeddings, similar to task-specific text embeddings [Su et al., 2022, Asai et al., 2022]. CLOC [Chen et al., 2024a] proposes the new learning goal of **promptable embeddings** for better localization given spatial hints. GiVE [Li et al., 2024] and FLAIR [Xiao et al., 2024] design patch-level or token-level image-text interaction mechanisms for language-informed image embeddings. Universal embedders [Wei et al., 2024, Zhang et al., 2024, Zhou et al., 2024a,b] leverage promptable query embeddings for the composed image retrieval task. For the retrieval targets, VLM2Vec [Jiang et al., 2024b] and E5-V [Jiang et al., 2024a] try using different texts as prompts when deriving image embeddings for some domains (e.g., news, fashion). MM-Embed [Lin et al., 2024] recognizes the benefit of prompts but requires fine-tuning with domain-specific instruction. Promptable embeddings are also applied to other areas like reinforcement learning [Chen et al., 2024b]. We systematically study the promptable image embeddings for retrieval targets and propose acceleration strategies.

**Localized Vision-Language Models (VLMs)** are motivated by similar ideas that some part of an image is of interest for a given task. V\* [Wu and Xie, 2023] formulates the problem as iterative visual search on a high-resolution image. Kosmos-2 [Peng et al., 2023] and GLIPv2 [Zhang et al., 2022] consider language grounding by bounding boxes or phrases. DAM [Lian et al., 2025] explores captioning for a given region. Lin et al. [2023] applies cropping to regions of interest, and Wang et al. [2024] aims at a similar task called Partial Scene Text Retrieval. The visual intelligence of OpenAI's o3 and o4-mini models is incorporated with simple tools like zooming and cropping to process the images for better reasoning [OpenAI, 2025], but their approach is region-based, while our attributes of interest could be non-region-based (e.g., `Scene` and `Time`).

## 3 Benchmarking T2I Retrievers on Attribute-Focused Queries

We focus on attribute-focused T2I queries in this work. In standard, general-purpose T2I benchmarks like MSCOCO [Lin et al., 2014], queries are image captions that describe the main content of the image (e.g., "`A black Honda motorcycle parked in front of a garage.`") but omitting other attributes like `Weather`, especially when they are non-dominant attributes. Hence, they cannot

Table 1: Model details of T2I retrievers and their average Recall@1 and Recall@5 (in percentage points) on the MSCOCO 2017 validation set and COCO-FACET. Recent MLLM-based universal embedders (shown in the second section) outperform CLIP-like models, but all T2I retrievers exhibit a performance drop on attribute-focused queries.

| Retriever | Img. Size | Params (M) | Output Dim. | COCO | | COCO-FACET | |
|---|---|---|---|---|---|---|---|
| | | | | Recall@1 | Recall@5 | Recall@1 | Recall@5 |
| CLIP-ViT-L/14 | $336^2$ | 427.9 | 768 | 81.0 | 97.9 | 33.7 | 47.0 |
| EVA01 ViT-g-14 | $224^2$ | 1136.4 | 1024 | 83.2 | 98.4 | 35.4 | 48.3 |
| EVA02 ViT-bigE-14+ | $224^2$ | 5044.9 | 1024 | 87.9 | **99.2** | 34.2 | 48.8 |
| SigLIP ViT-SO-14 | $384^2$ | 878.0 | 1152 | 73.2 | 95.5 | 37.8 | 51.9 |
| SigLIP2 ViT-SO-14 | $384^2$ | 1136.0 | 1152 | 87.4 | 98.4 | 39.8 | 52.6 |
| BLIP2-COCO | $224^2$ | 1173.2 | 768 | **88.8** | 99.1 | 37.8 | 51.6 |
| MagicLens | $224^2$ | 427.6 | 768 | 87.2 | **99.2** | **40.6** | **56.0** |
| E5-V | $336^2$ | 8355.3 | 4096 | 89.6 | 99.3 | 46.0 | 61.8 |
| MM-Embed | $336^2$ | 8175.5 | 4096 | 93.2 | **99.7** | 42.8 | 58.4 |
| MMRet-MLLM-S2 | $336^2$ | 7566.3 | 4096 | 93.7 | **99.7** | **48.8** | **64.5** |
| LLaVE-2B | $336^2$ | 1945.2 | 1536 | 92.4 | **99.7** | 45.6 | 59.5 |
| VLM2Vec-Phi-3.5-V | $336^2$ | 4146.6 | 3072 | 89.4 | 99.5 | 44.5 | 58.9 |

be used for our purpose directly. A recent benchmark MMVP-VLM [Tong et al., 2024] contains 270 fine-grained testcases but only has a small image pool (two images per query) designed for image-text matching instead of T2I retrieval.

Therefore, we construct a new benchmark, COCO-FACET, for evaluating T2I retrievers on attribute-focused queries in Section 3.1, and analyze their performance in Section 3.2. We reveal that current retrievers fall short of such queries, even though they involve less image-text matching than long MSCOCO-style captions with multiple attributes.

## 3.1 Benchmark Construction

We utilize the existing annotations provided by MSCOCO [Lin et al., 2014], Visual7W [Zhu et al., 2016], VisDial [Das et al., 2017], and COCO-Stuff [Caesar et al., 2018] about COCO images. In total, we collect 9,112 test cases covering eight types: 💼Objects, 🐦Animals, and 🤾Gestures of people in the image based on MSCOCO's annotations on segmentation, 🏖Scenes and 🕐Times of the day shown in the image based on Visual7W's where- and when-question answering, 👥Count of People in the image based on Visual7W's how-many-people-question answering, 🌦Weathers based on MSCOCO, Visual7W, and VisDial annotations, ⛏Materials of objects or surfaces shown in the images based on COCO-Stuff. While 💼Objects, 🐦Animals, and 🏖Scenes are extensively studied in areas like image classification, others are less-explored. Additionally, we consider both regional attributes (🐦Animals, 💼Objects, 🤾Gestures, ⛏Materials) and global attributes (🏖Scenes, 👥Count of People, 🕐Times, 🌦Weathers) which require inference based on the whole picture, so simple strategies like cropping or zooming may not be effective.

Each test case contains a text query (e.g., "Find me an everyday image that shows the scene of the beach."), a positive candidate (ground truth) that contains the attribute required in the query (the scene of the beach), and 99 negative candidates that do not contain such attribute. To ensure the quality of the negatives, we randomly select from images that have exclusive attributes to avoid ambiguity (e.g., images that show the scene of a conference room). More benchmark details and generation procedure are deferred to Appendix A and examples of each category are shown in Appendix D. We use the validation set of MSCOCO 2017 for comparison after converting it to the same format ("Find me an everyday image that matches the given caption."+COCO caption as the query text, and 100 candidate images).

## 3.2 Benchmark Results

We evaluate 12 state-of-the-art T2I retrievers and present the results in Table 1 and Table 2. The first seven rows feature the CLIP family (CLIP [Radford et al., 2021], EVA-CLIP [Fang et al., 2023, 2024], SigLIP [Zhai et al., 2023], SigLIP2 [Tschannen et al., 2025], and BLIP2 finetuned on COCO [Li et al.,

Table 2: Recall@1 and Recall@5 (in percentage points) for various text-to-image retrievers by category on our COCO-FACET benchmark (🐦: Animals, 🏖: Scenes, 💼: Objects, 👥: Count of People, 🛠: Materials, 🕐: Times, 🌦: Weathers, 🤸: Gestures). Cells shaded in red indicate low category-specific performance (Recall@1 < 10% or Recall@5 < 20%). All models struggle more on the last five attributes.

| | Img. Size | 🐦 | 🏖 | 💼 | 👥 | 🛠 | 🕐 | 🌦 | 🤸 |
|---|---|---|---|---|---|---|---|---|---|
| | | | | Recall@1 | | | | | |
| CLIP-ViT-L/14 | $336^2$ | 91.5 | 55.2 | 54.0 | 3.5 | 3.5 | 4.5 | 4.2 | 6.8 |
| EVA01 ViT-g-14 | $224^2$ | 93.7 | 56.4 | 58.1 | 2.3 | 2.8 | 4.6 | 2.9 | 7.0 |
| EVA02 ViT-bigE-14+ | $224^2$ | 88.1 | 55.8 | 55.9 | 3.3 | 3.1 | 4.6 | 3.2 | 7.1 |
| SigLIP ViT-SO-14 | $384^2$ | 92.4 | 57.0 | 63.1 | 4.4 | 3.2 | 4.7 | 3.8 | 7.5 |
| SigLIP2 ViT-SO-14 | $384^2$ | 94.9 | 54.1 | 66.2 | 4.9 | 3.4 | 3.6 | 4.8 | 11.2 |
| BLIP2-COCO | $224^2$ | 87.0 | 62.2 | 61.3 | 4.9 | 5.0 | 4.6 | 3.5 | 14.8 |
| MagicLens | $224^2$ | 94.7 | 70.9 | 63.2 | 15.9 | 6.2 | 10.0 | 3.2 | 14.3 |
| E5-V | $336^2$ | 92.7 | 70.4 | 71.2 | 31.4 | 10.5 | 7.5 | 5.2 | 20.1 |
| MM-Embed | $336^2$ | 92.7 | 67.4 | 68.1 | 13.3 | 7.4 | 5.1 | 3.9 | 19.5 |
| MMRet-MLLM-S2 | $336^2$ | 97.2 | 72.1 | 76.0 | 29.8 | 10.0 | 8.4 | 3.6 | 24.1 |
| LLaVE-2B | $336^2$ | 96.3 | 70.9 | 73.1 | 19.4 | 8.8 | 3.0 | 4.5 | 19.2 |
| VLM2Vec-Phi-3.5-V | $336^2$ | 95.5 | 69.8 | 74.4 | 14.4 | 6.3 | 5.5 | 4.3 | 9.8 |
| | | | | Recall@5 | | | | | |
| CLIP-ViT-L/14 | $336^2$ | 98.4 | 80.8 | 72.7 | 13.5 | 11.4 | 10.1 | 14.3 | 18.5 |
| EVA01 ViT-g-14 | $224^2$ | 99.0 | 84.9 | 75.6 | 12.1 | 9.6 | 12.4 | 12.8 | 19.1 |
| EVA02 ViT-bigE-14+ | $224^2$ | 98.7 | 83.7 | 75.5 | 15.9 | 10.8 | 12.6 | 14.6 | 19.9 |
| SigLIP ViT-SO-14 | $384^2$ | 99.5 | 81.4 | 80.3 | 13.3 | 14.5 | 21.5 | 13.1 | 20.8 |
| SigLIP2 ViT-SO-14 | $384^2$ | 99.3 | 79.1 | 82.2 | 18.7 | 12.3 | 12.4 | 13.5 | 25.2 |
| BLIP2-COCO | $224^2$ | 97.8 | 85.5 | 75.9 | 19.3 | 15.3 | 15.7 | 14.6 | 33.1 |
| MagicLens | $224^2$ | 99.3 | 89.5 | 81.3 | 35.7 | 15.7 | 25.1 | 16.3 | 32.1 |
| E5-V | $336^2$ | 98.3 | 91.9 | 89.0 | 60.1 | 25.2 | 18.6 | 16.1 | 35.5 |
| MM-Embed | $336^2$ | 98.7 | 87.8 | 84.9 | 29.3 | 20.3 | 15.0 | 15.3 | 45.3 |
| MMRet-MLLM-S2 | $336^2$ | 99.9 | 92.4 | 91.6 | 45.2 | 28.3 | 19.7 | 17.6 | 49.7 |
| LLaVE-2B | $336^2$ | 99.4 | 93.6 | 88.2 | 41.3 | 21.5 | 10.9 | 16.3 | 37.0 |
| VLM2Vec-Phi-3.5-V | $336^2$ | 99.3 | 90.7 | 90.7 | 36.6 | 18.2 | 12.9 | 19.2 | 27.1 |

2023b]) and MagicLens [Zhang et al., 2024], which use unimodal encoders. Although MagicLens accepts image+text as the input through a fusion module, we follow their T2I retrieval protocol and use the finetuned text and image encoders without other modules. As CLIP is sensitive to text format, we also conduct evaluation using standard CLIP-style texts ("a photo ...") for CLIP-ViT-L/14 in Appendix B.2 but find no significant difference. The last five rows are recent MLLM-based universal embedders that accept combinations of images and texts as input naturally: E5-V [Jiang et al., 2024a], MM-Embed [Lin et al., 2024], MMRet [Zhou et al., 2024a], LLaVE [Lan et al., 2025], and VLM2Vec [Jiang et al., 2024b]. Since our targets are image-only, we use the default, general texts (e.g., "Represent the given image." for VLM2Vec) accompanied with the images for encoding. More details about these models are in Appendix B.

**Current retrievers struggle on our benchmark.** Compared with Recall@1 and Recall@5 on original MSCOCO, the average performance on our benchmark degrades for all retrievers across different architectures and scales. In the first section, the performance among the CLIP family are close, with the most recent model, SigLIP2, scoring the highest on COCO-FACET. MagicLens has the best overall performance despite its small scale, output dimension, and image resolution. In the second section, universal multimodal embedders have better performance than unimodal embedders, with MMRet-MLLM-S2 in a relatively large scale outperforming others. Still, the performance differences are small among universal embedders, with a clear gap between COCO and COCO-FACET results.

**Retrievers have imbalanced performance on different attributes.** Again compared with COCO, models have lower performance on attributes apart from 🐦Animals, and significantly lower on the last five attributes in Table 2, indicating that they are largely neglected. Possible causes include

Table 3: Recall@1 and Recall@5 for text-to-image retrieval (in percentage points) on our COCO-FACET benchmark. Promptable image embeddings yield substantial average improvements and outperform the baselines on seven out of eight attributes.

| | 🕊 | 🏖 | 💼 | 👥 | 🛠 | 🕐 | 🌦 | 🏃 | Avg. |
|---|---|---|---|---|---|---|---|---|---|
| | | | | Recall@1 | | | | | |
| CLIP-ViT-L/14-336px | 90.9 | 55.2 | 53.1 | 3.5 | 3.5 | 4.5 | 4.2 | 6.8 | 33.7 |
| VLM2Vec-Phi-3.5-V | **95.5** | 69.8 | 74.4 | 14.4 | 6.3 | 5.5 | 4.3 | 9.8 | 44.5 |
| w/ GPT prompt | 90.7 | **81.4** | **75.5** | **72.7** | **25.8** | **18.4** | **14.4** | **15.7** | **53.4** |
| Text-Based | 69.9 | 66.9 | 62.6 | 35.6 | 13.5 | 13.2 | 7.1 | 10.7 | 40.5 |
| | | | | Recall@5 | | | | | |
| CLIP-ViT-L/14-336px | 97.9 | 80.8 | 72.2 | 13.5 | 11.4 | 10.1 | 14.3 | 18.5 | 47.0 |
| VLM2Vec-Phi-3.5-V | **99.3** | 90.7 | 90.7 | 36.6 | 18.2 | 12.9 | 19.2 | 27.1 | 58.9 |
| w/ GPT prompt | 98.7 | **95.9** | **92.0** | **92.1** | **48.8** | **82.4** | **36.5** | **39.3** | **75.5** |
| Text-Based | 90.7 | 86.6 | 81.7 | 60.6 | 37.2 | 39.2 | 24.4 | 26.1 | 60.6 |

reporting bias [Kamath et al.]—"people murder" is more likely to appear than "people breathe" in the corpora, as in our case the Time (e.g., morning or night) might be too obvious to report—in both training data and previous evaluation. Meanwhile, this verifies the findings of prior work on visual grounding and reasoning [Paiss et al., 2023, Chen et al., 2024a, Tong et al., 2024] in the setting of T2I retrieval, that CLIP-like models only achieve global text-image alignment, and thus their image embeddings focus on global semantics or subjects while leaving out attributes like object details and quantity.

**Dominant-but-wrong attributes may be favored over correct-but-non-dominant ones.** We look into the failure cases of CLIP-ViT-L/14 and VLM2Vec as representative retrievers. As shown in Figure 1, they seem to rank higher the images with a similar attribute (a motorcycle or a train) as the main content but not the one with the correct attribute as non-dominant elements (a car in the background). More examples can be found in Appendix D. This implies that "simple" images consisting of fewer, more salient attributes may be preferred in retrieval rather than "complex" images at the cost of precision.

## 4 Promptable Image Embeddings

When compressing images into embeddings of limited length, some visual information may be discarded, leading to low performance on relevant queries. Hence, a natural approach to address this issue is to (1) learning a denser visual representation during pretraining, or (2) using embeddings with larger dimensions. However, recent SigLIP2 [Tschannen et al., 2025] trained with a global-local loss for improving fine-grained local semantics only slightly outperforms other CLIP-like models, and using 4096-dimensional large embeddings (like in MMRet-MLLM-S2) still fails to resolve this issue, with Recall@1 lower than 10% and Recall@5 lower than 20% for some categories. Based on these findings, we hypothesize that *general* image embeddings might be inefficient for attribute-specific queries.

We therefore focus on highlighting the important part in the embeddings. For this purpose, we propose to use **promptable image embeddings** for different attributes—conditioning image embeddings on textual prompts, which is enabled by recent MLLM-based universal embedders. Although they are mainly motivated by tasks involving combinations of images and texts as queries or targets (e.g., composed image retrieval), previous research [Jiang et al., 2024a] finds that they can (1) deal with unseen prompts since they are based on pre-trained MLLMs, and (2) accept task-specific prompts accompanied with images, such as modification in the FashionIQ dataset [Wu et al., 2021]. In Section 4.1, we formally study the promptable image embeddings and demonstrate that our designed pipeline yields a performance boost for attribute-focused queries. Moreover, we show that it surpasses text-based T2I retrieval (Section 4.2).

### 4.1 Method

We employ VLM2Vec-Phi-3.5-V [Jiang et al., 2024b] as the base retriever for deriving promptable image embeddings in this subsection. This model is built on Phi-3.5-V [Abdin et al., 2024], which

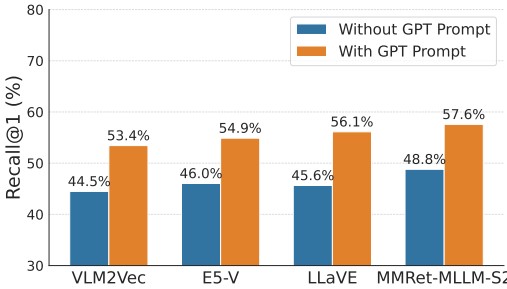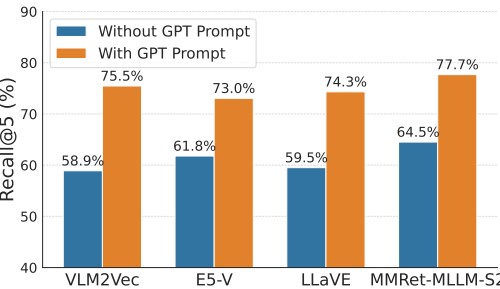

Figure 2: Average retrieval performance across various base retrievers on COCO-FACET, with and without GPT-generated prompts. The same set of prompts brings consistent performance gain on different multimodal base retrievers.

has strong capabilities in image understanding. VLM2Vec generates embeddings by taking the last layer vector representation of the last token as the embedding and is fine-tuned with a contrastive loss. Hence, when given a piece of text and an image, it produces a single combined embedding.

The authors previously tried using question+image as the input for embedding queries for the VQA tasks, while the targets are the ground-truth, text-only answers. This motivates us to use questions as prompts for target images to ask about the required attributes in our case. Ideally, the resulting embeddings would contain "answers" to such constructed visual questions, which is the corresponding attribute of the image. To make the pipeline automatic and extendable, we use GPT-4o [Hurst et al., 2024] to generate questions with the following template:

```
Write a question to ask about the {Attribute Name} in a image, with possible
answers such as {A}, {B}, and so on.  Please answer in one sentence without
mentioning any answer.
```

For example, the output for input tuple (people gesture, standing, jumping) is "What gesture are the people making in this image?" Then we concatenate this question with the fixed template of the base retriever ("<|image_1|> Represent the given image with the following question:") to construct the full prompt used for model input. **We show that the promptable image embeddings improve the performance of VLM2Vec-Phi-3.5-V, especially on the five more challenging categories, by highlighting the corresponding attributes** (See Table 3). In addition, we test human-written prompts and observe a similar performance gain, and attach all prompts in Appendix B.3. We further evaluate a recent non-MLLM-based retriever with language-informed image representation for highlighting relevant patches but find little improvement (FLAIR [Xiao et al., 2024] in Appendix C.2). This underlines the advantage of inheriting the image-text alignment in pretrained MLLMs with a flexible image focus.

**Furthermore, the same set of prompts generalizes to other universal multimodal embedders used as base retrievers** (E5-V, LLaVE, MMRet-MLLM-S2) (See Figure 2). We also apply them to MM-Embed, but this model lacks the zero-shot ability on prompting and requires fine-tuning as mentioned in their paper. See the detailed numerical results by category in Appendix C.1.

**Additionally, the method works on different image pools.** We converted two scene classification benchmarks, Place365 [Zhou et al., 2017] and SUN397 [Xiao et al., 2010] provided in MMEB with 1,000 test cases for each [Jiang et al., 2024b], into text-to-image retrieval benchmarks in the same format, except that we use one positive candidate and 499 negative candidates for each test case. We notice that the same prompt used for 🏖Scene increases both Recall@1 and Recall@5 performance in Table 4.

Table 4: Recall@1 and Recall@5 of T2I retrieval for the converted Place365 and SUN397, comparing promptable image embeddings with original image embeddings. The improvement indicates that our pipeline generalizes effectively to these image pools.

|  | Place365 | SUN397 |
|---|---|---|
| Recall@1 | | |
| VLM2Vec-Phi-3.5-V | 30.0 | 57.0 |
| w/ GPT prompt | **33.9** | **61.9** |
| Recall@5 | | |
| VLM2Vec-Phi-3.5-V | 58.2 | 85.2 |
| w/ GPT prompt | **66.7** | **88.9** |

## 4.2 Comparison with Text-Based T2I Retrieval

As we mention in Section 4.1, the promptable image embeddings are likely to contain the corresponding attributes of the image as "answers" to the prompt. Another similar approach is to directly ask an MLLM the same question about the image and to obtain a pure-text answer. Then, we can use the text embedding of this answer as the target embedding, which is explored in previous work [Karthik et al., 2023]. The pure-text answer is also supposed to be a dense representation of the image which contains required attributes in the prompt.

Is this text-based approach equivalent to our method? We conduct such comparison using LLaVA-1.5 [Liu et al., 2024] as the MLLM and GRIT [Muennighoff et al., 2024] as the text embedder. As shown in the last row in Table 3, this text-based approach underperforms our method, and it even loses to VLM2Vec-Phi-3.5-V without prompt in Recall@1. When checking its failure cases, we find that it suffers a lot from hallucination: For instance, when asking LLaVA-1.5 about animals' existence in an image with only feathers in a container, it wrongly answers with "there are birds visible." Besides, it cannot process the linguistic ambiguity in a pure-text answer (e.g., "chicken" can refer to a domestic animal or a type of meat). Some examples are attached in Appendix C.5. This indicates that the promptable image embedding provides more than an embedding of the pure-text answer to the given visual question in the prompt.

## 5 Acceleration

While effective, the pipeline has high computational cost in real-world T2I retrieval. For the experiments in Section 4.1, we assume that the query type is known during the pre-processing stage for computing the promptable image embeddings. However, if the query type is only known at test time, we need to take the per-query computational cost into consideration. Assume that we have $N$ images in the pool and $M$ text queries. Since $N$ is typically large, the ideal per-query cost should not grow linearly with $N$.

We focus primarily on embedding cost, as the maximum cosine-similarity searching step can be efficiently handled by the FAISS library [Douze et al., 2024]. Let $v$ denote the cost of computing a single image embedding, and $t$ the cost for a single text embedding, using CLIP-like models. Let $F$ represent the cost of a single forward pass through the base model of our multimodal embedder (e.g., Phi-3.5-V with a CLIP vision encoder). For CLIP-like models, the total embedding cost is $Nv + Mt$, with a per-query embedding cost of $t$. If we stick to the original pipeline for promptable image embeddings, the total embedding cost will be $Nv + M(NF + F)$, leading to a per-query cost of $NF + F$.

To reduce the per-query cost, we explore two strategies on VLM2Vec-Phi-3.5-V in this section. (1) The first approach is straightforward: We predefine potentially useful prompts and pre-process the promptable image embeddings. (2) The second solution is to derive linear approximation of the retriever at test time. Both strategies have lower per-query embedding cost and outperform the baseline VLM2Vec-Phi-3.5-V on Recall@5 of COCO-FACET.

### 5.1 Pre-Processing Embeddings

Since many attributes of interest can be predicted beforehand with prior knowledge of the incoming tasks, we can predefine some prompts at the pre-processing stage using our pipeline. During inference, we only need to select the most suitable prompt and retrieve from the corresponding promptable image embeddings.

We test this strategy on the COCO-FACET benchmark using the prompt set obtained in Section 4.1. We use GPT-4o for prompt selection at test time with the template attached in Appendix B.4. The ground-truth prompt for each query can be selected with high accuracy on average at test time. The low selection accuracy for the 🤾Gesture category is due to the similar prompt in 👥Count of People category, but we find that such a similar prompt other than the ground truth could also lead to correct answers, indicating some degree of error tolerance. (See Table 5.)

The per-query embedding cost of this strategy is $F$ (for embedding the query text), and there is an additional cost for calling the GPT-4o API. The $NF$ term in the embedding cost is replaced by a higher memory cost and pre-processing time cost.

Table 5: Recall@1 and Recall@5 (in percentage points) of accelerated text-to-image retrieval with pre-processed promptable image embeddings on the COCO-FACET benchmark. The ground-truth prompt can be selected with a high accuracy for most categories.

| | 🕊️ | ⛱️ | 💼 | 👥 | 🛠️ | 🕐 | 🌦️ | 🤺 | Avg. |
|---|---|---|---|---|---|---|---|---|---|
| | | | | Recall@1 | | | | | |
| VLM2Vec-Phi-3.5-V | **95.5** | 69.8 | 74.4 | 14.4 | 6.3 | 5.5 | 4.3 | 9.8 | 44.5 |
| w/ selected GPT prompt | 90.9 | **81.4** | **75.5** | **72.7** | **25.8** | **18.4** | **14.4** | **10.5** | **52.8** |
| w/ gt GPT prompt | 90.7 | 81.4 | 75.5 | 72.7 | 25.8 | 18.4 | 14.4 | 15.7 | 53.4 |
| | | | | Recall@5 | | | | | |
| VLM2Vec-Phi-3.5-V | **99.3** | 90.7 | 90.7 | 36.6 | 18.2 | 12.9 | 19.2 | **27.1** | 58.9 |
| w/ selected GPT prompt | 99.1 | **95.9** | **92.0** | **92.1** | **48.8** | **82.4** | **36.5** | 25.3 | **73.7** |
| w/ gt GPT prompt | 98.7 | 95.9 | 92.0 | 92.1 | 48.8 | 82.4 | 36.5 | 39.3 | 75.5 |
| Selection Acc. | 100 | 86.6 | 99.9 | 100 | 100 | 100 | 100 | 8.9 | 87.9 |

Table 6: Recall@1 and Recall@5 (in percentage points) of accelerated text-to-image retrieval with approximated promptable image embeddings ($K = 100$) on the COCO-FACET benchmark. Results are averaged over five independent runs.

| | 🕊️ | ⛱️ | 💼 | 👥 | 🛠️ | 🕐 | 🌦️ | 🤺 | Avg. |
|---|---|---|---|---|---|---|---|---|---|
| | | | | Recall@1 | | | | | |
| VLM2Vec-Phi-3.5-V | **95.5** | 69.8 | **74.4** | 14.4 | 6.3 | 5.5 | 4.3 | 9.8 | **44.5** |
| w/ linear approx. | 72.1 | 67.2 | 57.1 | **47.5** | **24.3** | **35.7** | **9.0** | **14.2** | 42.5 |
| w/ GPT prompt | 90.7 | 81.4 | 75.5 | 72.7 | 25.8 | 18.4 | 14.4 | 15.7 | 53.4 |
| | | | | Recall@5 | | | | | |
| VLM2Vec-Phi-3.5-V | **99.3** | 90.7 | **90.7** | 36.6 | 18.2 | 12.9 | 19.2 | 27.1 | 58.9 |
| w/ linear approx. | 84.6 | **91.9** | 83.2 | **73.7** | **43.9** | **71.5** | **28.4** | **38.1** | **67.0** |
| w/ GPT prompt | 98.7 | 95.9 | 92.0 | 92.1 | 48.8 | 82.4 | 36.5 | 39.3 | 75.5 |

## 5.2 Linear Approximation at Test Time

When there are novel attributes required in the query, can we process it with a lower cost? We can first use the previous automatic pipeline to get a prompt $p$. Assuming that the model has access to images in the same category as the query, we experiment with a test-time linear approximation of the universal embedder[1]. Specifically, we denote the normalized original image embeddings without a prompt as $a$, and the normalized promptable image embeddings as $b$. Let $U$ be the multimodal embedder and $q$ be the normalized query embedding. We would like to find a matrix $W$ with respect to $p$ such that

$$Wa \approx U(a, p) = b$$

for all $a$ in the image pool. After deriving $W$, we can use $Wa$ for retrieval, searching for the $a$ such that the dot product between $Wa$ and $q$ is maximized.

We find this matrix $W$ based on a small amount of $(a, b)$ pairs. During pre-processing, we store $a$ for all images in the pool. At test time, after obtaining $p$, we uniformly sample $K$ images from the candidate pools of all queries in the same category along with their $a$ and calculate $U(a, p) = b$, denoting by matrices $A$ and $B$. The best linear approximation is then given by $W = BA^\top$. $W$ can be applied to the query $q$ instead, since $(Wa)^\top q = a^\top(W^\top q)$. The per-query embedding cost is thus $KF + F$, while the pre-processing cost remains unchanged. It is worth noting that theoretically, $W$ should be orthogonal to ensure that $||Wa||_2 = 1$ as $||b||_2 = 1$, but in practice we find that the $W$ directly derived from $BA^\top$ and normalized after being applied to $q$ works well. Thus, strict orthogonality is not required.

We test the method on VLM2Vec-Phi-3.5-V with $K = 100$ for each category. The results are shown in Table 6, with the error bar listed in Appendix C.3. Although the linear approximation of a MLLM-based embedder has limited expressiveness and cannot capture the nonlinear, complex

---

[1]We include experiments that ablate this assumption in Appendix E.

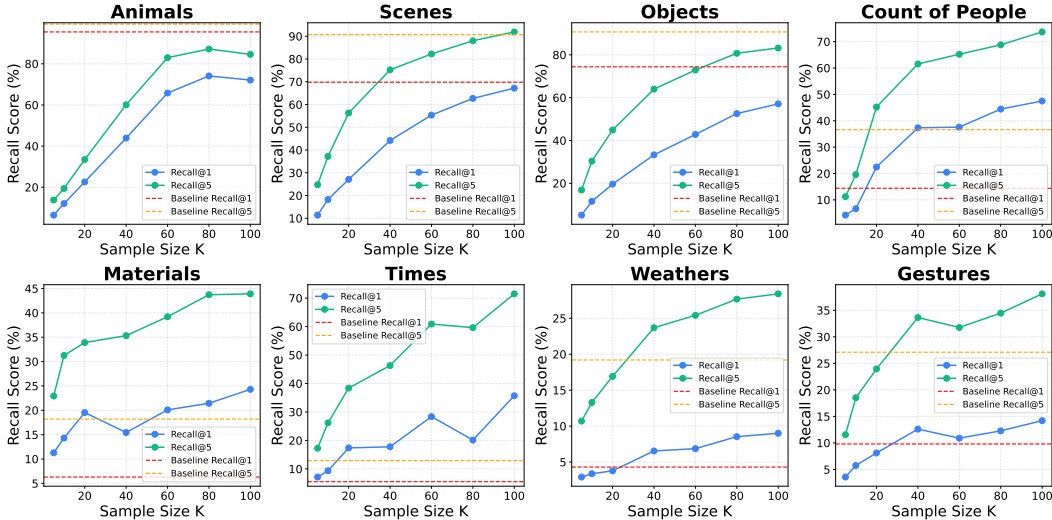

Figure 3: Recall@1 and Recall@5 of accelerated text-to-image retrieval using approximated prompt-able image embeddings with varying sample size $K$ in percentage points on COCO-FACET. The results are averaged over five independent runs. "Baseline" refers to using VLM2Vec-Phi-3.5-V without prompts.

cross-modal interactions, it still improves the baseline on the harder attributes and yields a better Recall@5.

We also test smaller $K$ for cost-flexible deployment. When we vary $K$ from 5 to 100, both metrics show overall improvement in Figure 3, allowing tuning $K$ based on available compute and desired latency. On ⚒Materials and 🕐Times, using a very small sample size ($K = 5$) outperforms the baseline model without prompt. On other harder attributes, $K = 40$ is sufficient for exceeding the baseline. However, it is still challenging to match the baseline on the three more common attributes (💼Animals, 🏔Scenes, and 💼Objects), which is a limitation of current approximation approach. We defer the actual cost analysis to Appendix C.3 and visualization to Appendix C.4.

## 6 Conclusion

We introduced COCO-FACET, a benchmark to evaluate text-to-image retrieval performance on attribute-focused queries, revealing limitations in current CLIP-like and MLLM-based retrievers. General-purpose image embeddings often overlook fine-grained visual attributes critical for accurate retrieval. To address this, we propose to use promptable image embeddings on MLLM-based universal embedders, which improve focus on relevant attributes and enhance retrieval quality while being flexible, model-agnostic. We explore efficient acceleration strategies that makes it more practical for deployment. Together, our work offers a promising direction for building more precise and efficient T2I retrievers that can be integrated into systems for general problem-solving.

## Acknowledgments and Disclosure of Funding

SSD acknowledges the support of NSF DMS 2134106, NSF IIS 2143493, NSF IIS 2229881, Sloan Fellowship, and the AI2050 program at Schmidt Sciences.

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

# A  The COCO-FACET Benchmark

We provide more details on the COCO-FACET Benchmark in this section.

Building upon the annotations of MSCOCO 2017 captions [Lin et al., 2014], COCO-Stuff [Caesar et al., 2018], Visual7W [Zhu et al., 2016], and Visual Dialog [Das et al., 2017] based on the COCO images, we generate 8 new subsets for evaluating T2I retrievers on attribute-focused retrieval tasks. Use of the COCO images must abide by the COCO anf Flickr Terms of Use[2]. The MSCOCO 2017 and COCO-Stuff annotations belong to the COCO Consortium and are licensed under a Creative Commons Attribution 4.0 License. The Visual7W annotations are under MIT License. The Visual Dialog annotations are licensed under a Creative Commons Attribution 4.0 International License.

We extract images with different types of objects including *laptops*, *bicycles* and *sandwiches* on it, with 3749 test cases in total; 8 gestures of people including *standing* and *sitting* on it, with 1176 test cases in total; and other 6 benchmarks showing the other attributes. The details of the dataset construction is introduced in Appendix A.1, where the full list of categories and related statistics can be seen in Table 7.

In short, we came up with the most proper division for every image in every fixed criteria. The information needed for the classification might come from the whole picture or a part of it. Each test case consists of a query text (e.g., "Find me an everyday image showing some object or surface made of stone"), an image corresponding to the text (e.g. a photo of a stone wall) and 99 negatives that should be divided into other categories with no intersection with the true category in the classification for the retrieval (e.g. a photo of a wooden floor). For each of the test cases in these provided data, the chance level performance to choose a correct image is 1%.

## A.1  Generating Datasets

We use these steps as a norm when generating the datasets. Details for each benchmark will be shown in Appendix A.2.

1. We find a proper criteria that can include sufficient images and categories.

2. Initially, we filter the captions or dialogues within the datasets to identify those containing information in specific categories.

3. Then we analyze some filtered samples and refine the rules of filtering.

4. We remove the samples that belong to multiple mutually exclusive categories under the rule.

5. Subsequently, we manually validate the positive images to minimize wrong labels and ambiguity.

6. We design the queries and randomly select negative samples from mutually exclusive categories.

## A.2  Details of Sets

See Table 7 for the category statistics.

### A.2.1  Gestures of People

- Idea: The position and arrangement of different parts of the human body vary significantly depending on the gesture, such as standing, sitting, or lying. Therefore, a classification based on gestures should be feasible.

- Categories: "stand", "sit", "jump", "lie", "bend", "squat", "kneel", "crawl" (8 categories, 1176 samples in total).

- Avoidance: We avoid using "squat" or "kneel" as negative samples for each other, as they can be hard to distinguish (e.g., a kneeling squat).

- Sampling: All samples were sourced from the val2017 split of the COCO dataset. We first filter captions using keywords such as "stand" or "sitting." Then, we refine the keywords

---

[2] http://cocodataset.org/#termsofuse and https://www.flickr.com/creativecommons/

Table 7: Statistics of our COCO-FACET benchmark(🕊: Animals, 🏖: Scenes, 💼: Objects, 👥: Count of People, 🛠: Materials, 🕐: Times, 🌥: Weathers, 🤸: Gestures).

| | NUM. Samples | NUM. Categories | Details of each category |
|---|---|---|---|
| 🕊 | 763 | 10 | "bird": 44, "cat": 141, "dog": 112, "horse": 88, "sheep": 48, "cow": 56, "elephant": 79, "bear": 42, "zebra": 73, "giraffe": 80 |
| 🏖 | 172 | >90 | omitted |
| 💼 | 3849 | 69 | omitted |
| 👥 | 570 | 12 | "0": 131, "1": 88, "2": 106, "3": 86, "4": 42, "5": 22, "6": 23, "7": 11, "8": 8, "9": 8, "10": 7, "over 10": 38 |
| 🛠 | 1128 | 5 | "wood": 231, "stone": 57, "metal": 741, "paper": 68, "brick": 31 |
| 🕐 | 760 | 7 | "daytime": 588, "night": 103, "afternoon": 24, "dusk": 3, "morning": 15, "sunrise": 1, "evening": 26 |
| 🌥 | 694 | 12 | "sunny": 179, "clear": 77, "misty": 14, "overcast": 17, "cloudy": 138, "rainy": 50, "drizzly": 1, "stormy": 2, "snowy": 193, "warm": 8, "cold": 14, "chilly": 1 |
| 🤸 | 1176 | 8 | "stand": 660, "sit": 386, "lie": 25, "jump": 77, "bend": 10, "squat": 3, "kneel": 14, "crawl": 1 |

with more precise phrases to avoid mislabeling images based on captions like "A building sits between." Finally, we manually check the images to ensure that they meet our criteria.

### A.2.2 Materials of Objects or Surfaces

- Idea: Although different materials may have similar uses, they often exhibit different visual characteristics. Therefore, a classifier should be able to distinguish between them.
- Categories: "wood", "metal", "stone", "brick", "paper" (5 categories, 1128 samples in total).
- Avoidance: We avoid using "stone" or "brick" as negative samples for each other. This is because some images contain bricks made of stone. Since we categorize such images under "stone", we aim to avoid interfering with the classification of "brick".
- Sampling: We utilized the COCO-Stuff annotations on val2017, extracting subcategories that fall under our target categories. Then, we manually review the images and exclude samples that may contain multiple materials, have unclear material identification, or only show a small portion of the target material.

### A.2.3 Count of People

- Idea: Object detection models are capable of accurately locating object boundaries. Therefore, it is reasonable to infer that they can accurately count the number of people in an image.
- Categories: "0", "1", "2", "3", "4", "5", "6", "7", "8", "9", "10", "over 10" (12 categories, 570 samples in total).
- Avoidance: We require that the negative samples differ from the positive sample by at least 3. This accounts for potential ambiguity regarding whether certain individuals should be

included or excluded, providing a margin for a potential count discrepancy of one person in both positive and negative samples. Specifically, "over 10" is treated as 11 when calculating the differences.

- Sampling: We primarily use the Visual7W VQA dataset as our resource. We first filter questions that contain interrogative phrases like "how many people" Then, we select questions with clear answers relevant to our task, either manually verify them or deduce the final answer from similar questions (e.g., "how many people other than a person") and visually confirm.

### A.2.4 Weather Conditions

- Idea: Weather conditions are relatively easy to distinguish in images showing a large outdoor scene. In addition, there are many different criteria for evaluating weather, allowing a substantial number of categories.

- Categories: "sunny", "clear", "misty", "overcast", "cloudy", "rainy", "drizzly", "stormy", "snowy", "warm", "cold", "chilly" (12 categories, 694 samples in total).

- Avoidance: For the first two categories ("sunny" and "clear"), we avoid using them as negative samples for each other because they both describe a sky with no clouds. We also avoid using "warm" or "sunny" as negative samples as these conditions can coexist with a clear or sunny sky.

  Categories three through eight ("misty" to "stormy") are not used as negative samples for each other, as they all describe conditions involving clouds, rain, or something that obstructs sunlight, making them difficult to delineate clearly. Furthermore, we did not use "cold" or "chilly" as negative samples for these conditions, as it was difficult to determine the precise temperature under such circumstances. Categories three through five ("misty", "overcast", "cloudy") are not used as negative samples with "warm" for similar reasons (We assume that it is generally not warm when it is raining).

  Among the last four categories ("snowy", "warm", "cold", "chilly"), "snowy", "cold", and "chilly" often co-occur and are therefore not used as positive and negative samples together.

- Sampling: We first extract questions containing weather conditions from the Visual7W and VisDial datasets. Then, we record, simplify and verify the answers. In addition, we deduce weather conditions from the captions in val2017 and perform a double check.

### A.2.5 Time of the Scenery

- Idea: The Visual 7W dataset includes "When" as a category of questions. These questions often have a clear answer, thus we can get the time of the scene. This is a proper benchmark, since time is a property of the whole image.

- Categories: "daytime", "night", "evening", "afternoon", "dusk", "morning", "sunrise"(7 categories, 760 samples in total)

- Avoidance: We avoid using samples from a similar category as negative samples. For instance, "morning" is not a good negative sample for "sunrise".

- Sampling: We filter the answers of the time questions in Visual 7W. The core word is used as our category.

### A.2.6 Scenes of the Locations or Activities

- Idea: Different locations and activities have quite distinct scenes. Through the"where" questions of the Visual 7W set, we can easily get a description of the scene (a noun), and repetitions hardly exist.

- Categories: "beach (scene)", "beach shore", "baseball game", "Oahu", "baseball field", "baseball park", "sports arena", "outdoor eating area", "bedroom", "bathroom", "station", "train station", "railroad tracks", "backyard", "backyard patio", "zoo", "tennis court", "Broadway", "harbor", "street", "city street", "side of the road", "mountain", "river", "safari", "grassland", "airport", "air strip", "parking lot(area)", "skate park", "open field", "field", "construction site", "classroom", "fountain", "London", "nature", "farm", "restaurant", "dinner", "dining room", "kitchen", "kitchen being remodeled in a home", "living room", "park",

"ski slope", "(ski) lodge", "sidewalk", "parlor", "boardwalk", "waterhole", "baby shower", "press conference", "apple computers", "downtown Toronto", "outside a city", "near a river", "by/near the ocean", "ocean shore", "inside a home", "in a room", "inside a refrigerator", "near the food", "performance", "market", "farmers market", "a man on a phone in a room", "tourist trap", "coffee shop", "on a desk", "table", "on a counter", "sky", "woods", "birthday party", "outdoor show", "yard", "soccer field", "indoors", "in front of clock tower", "on a road in front of a large building", "in a building", "in front of a television", "third street", "in a car", "airport runway", "intersection", "museum", "concert photography session", "inside a home very close to a marina and the sea", "road", "Tokyo", "by the water", "on sand dune", "bakery", "motorcycle race", "house"(172 samples in total)

- Avoidance: We just avoid the categories with similar meaning to be used as both positive and negative. In fact, we manually check all the negative samples to avoid conflicting with the positive sample.

- Sampling: We used the "Where" questions from the Visual 7W set. The answer was often used directly as the category to expand the number of categories.

### A.2.7 Objects contained

- Idea: The COCO dataset provides the retrieval for objects in it, and the position of the objects are located with bounding boxes. So whether the object is a crucial feature in the picture is really clear. To be clear, we do not involve animals or people in this classification, for they have totally different norms.

- Categories: "bicycle", "bus", "light", "backpack", etc. (3849 samples in total)

- Avoidance: The COCO set gives quite comprehensive content about the objects in the image. So we just need to avoid using the images with the same kind of objects (possibly with other objects) as negative samples.

- Sampling: We can just use the annotations of objects for the COCO validation set.

### A.2.8 Animals Contained

- Idea: Animals have a great difference from static objects; they are classified with their appearances and actions. As a result, we create an independent benchmark.

- Categories: "giraffe", "zebra", "bear", "elephant", "cow", "sheep", "horse", "dog", "cat", "bird" (10 categories, 763 samples in total)

- Avoidance: We just need to avoid the same species or some close species in the negative samples.

- Sampling: We include the objects marked in the COCO validation set that are animals or similar to animals.

## B  Details of Retrievers

We list the details of retrievers used in our evaluation in this section, including the baselines and the promptable image embeddings. The evaluation code is attached in the supplementary material for reproducibility purpose.

### B.1  Information of Baselines

All evaluations of CLIP-family, MagicLens, MLLM-based universal multimodal retrievers, and variants of VLM2Vec can be done using one A6000 GPU with 48GB memory in less than 6 hours per category.

**CLIP-family:** The CLIP family comprises vision-language models trained via contrastive learning on large-scale image-text pairs. CLIP [Radford et al., 2021] introduced this paradigm, enabling zero-shot transfer to various vision tasks. We use the weights at `https://huggingface.co/openai/clip-vit-large-patch14-336` under MIT License. EVA-CLIP [Fang et al., 2023, 2024] enhances CLIP by integrating improved training techniques for better efficiency and effectiveness. We access their public model weights at OpenCLIP [Ilharco et al., 2021] with

model name "EVA01-g-14" and "EVA02-E-14-plus" under MIT license. SigLIP [Zhai et al., 2023] replaces the softmax loss with a sigmoid loss, allowing for scalable training without the need for large batch sizes. Building upon this, SigLIP2 [Tschannen et al., 2025] incorporates multilingual capabilities and improved semantic understanding. We use the weights at `https://huggingface.co/google/siglip-so400m-patch14-384` for SigLIP and `https://huggingface.co/google/siglip2-so400m-patch14-384` for SigLIP2 under Apache license 2.0. BLIP2 [Li et al., 2023b] fine-tuned on COCO leverages a frozen image encoder and a lightweight Q-Former to bridge vision and language modalities effectively. We use the "blip2_feature_extractor" provided by LAVIS [Li et al., 2023a] under BSD 3-Clause License. By default, we directly use the query text as text input for these models.

**MagicLens:** MagicLens [Zhang et al., 2024] is a self-supervised image retrieval model trained on 36.7M triplets of (query image, instruction, target image). It supports open-ended instructions, enabling retrieval based on diverse semantic relations beyond visual similarity. The model employs a dual-encoder architecture with shared parameters and utilizes multi-head attention pooling to generate unified embeddings. We use the weights shared in the official github repository at `https://github.com/google-deepmind/magiclens` under Apache-2.0 license. We only use their vision encoder and language encoder like CLIP, as we find that the model does not support zero-shot instructions for embeddings.

**E5-V:** E5-V [Jiang et al., 2024a] adapts an MLLM to generate universal multimodal embeddings. Unlike traditional models trained on image-text pairs, E5-V leverages MLLM's capabilities to represent multimodal information effectively, demonstrating significant potential in various retrieval tasks. We use the model weights released at `https://huggingface.co/royokong/e5-v`.

**MM-Embed:** MM-Embed [Lin et al., 2024] is a universal multimodal retrieval model that fine-tunes MLLMs as bi-encoder retrievers across diverse datasets and tasks. It supports flexible vision-language alignment and is adaptable to both retrieval and classification tasks without the need for instruction tuning. However, we find that the model does not process zero-shot instructions well. We use the weights at `https://huggingface.co/nvidia/MM-Embed` under Creative Commons Attribution Non Commercial 4.0.

**MMRet:** MMRet [Zhou et al., 2024a] is trained on MegaPairs, a massive synthetic dataset generated using vision-language models and open-domain images. It employs separate encoders for vision and language, followed by deep fusion layers for cross-modal alignment, achieving state-of-the-art performance in universal multimodal retrieval tasks. We employ the MMRet-MLLM-S2 released at `https://huggingface.co/BAAI/BGE-VL-MLLM-S2` under the MIT license.

**LLaVE:** LLaVE [Lan et al., 2025] introduces hardness-weighted contrastive learning to train large language and vision embedding models. By dynamically adjusting the learning process based on the difficulty of negative pairs, LLaVE enhances representation learning, leading to improved performance across various multimodal tasks. We use the LLaVE-2B released at `https://huggingface.co/zhibinlan/LLaVE-2B` under Apache license 2.0.

**VLM2Vec:** VLM2Vec [Jiang et al., 2024b] transforms vision-language models into efficient multimodal embedders through contrastive training on the Massive Multimodal Embedding Benchmark (MMEB). It supports instruction-guided representation generation, outperforming existing models on both in-distribution and out-of-distribution datasets. We use the VLM2Vec-Phi-3.5-V at `https://huggingface.co/TIGER-Lab/VLM2Vec-Full` under Apache license 2.0.

## B.2 Retrieval with CLIP-Style Text

Our query text is designed to suit the retrieval tasks in universal multimodal embedders such as VLM2Vec. So, a question arises when we try to evaluate with CLIP, in which the recommended CLIP evaluation starts with "A photo of." We perform an ablation study on replacing the text with CLIP-style text for evaluation. The mechanism of substitution is shown in Table 8. The results are shown in Table 9, where no significant difference is observed.

Table 8: CLIP-style text used in our evaluation.

| Original Text | Revised Text | Examples |
|---|---|---|
| "Find me an everyday image that···" | "A photo that···" | "Find me an everyday image that is taken during the evening." → "A photo that is taken during the evening." |
| "Find me an everyday image with···" | "A photo with···" | "Find me an everyday image with over 10 people." → "A photo with over 10 people." |
| "Find me an everyday image showing···" | "A photo showing···" | "Find me an everyday image showing some object or surface made of brick." → "A photo showing some object or surface made of brick." |

Table 9: Recall@1 and Recall@5 of CLIP-ViT-L/14's evaluation with original text or with CLIP-style text in percentage points on our COCO-FACET benchmark.

| | 🕊 | ⛱ | 💼 | 👥 | 🛠 | 🕐 | 🌦 | 🤸 | Avg. |
|---|---|---|---|---|---|---|---|---|---|
| | | | | Recall@1 | | | | | |
| CLIP-ViT-L/14 | 91.5 | 55.2 | 54.0 | 3.5 | 3.5 | 4.5 | 4.2 | 6.8 | 33.7 |
| w/ CLIP-style text | 91.5 | 53.5 | 54.0 | 8.4 | 5.1 | 4.5 | 2.6 | 4.8 | 33.8 |
| | | | | Recall@5 | | | | | |
| CLIP-ViT-L/14 | 98.4 | 80.8 | 72.7 | 13.5 | 11.4 | 10.1 | 14.3 | 18.5 | 47.0 |
| w/ CLIP-style text | 98.4 | 79.1 | 72.7 | 13.3 | 14.4 | 9.9 | 16.4 | 19.1 | 47.6 |

Table 10: GPT-written prompts for COCO-FACET.

| Categories | Prompts |
|---|---|
| 🕊 Animals | `<|image_1|> Represent the given image with the following question:  Which animals can be seen in this image?` |
| ⛱ Scenes | `<|image_1|> Represent the given image with the following question:  What type of location is depicted in this image?` |
| 💼 Objects | `<|image_1|> Represent the given image with the following question:  Which objects are present in this image?` |
| 👥 Count of People | `<|image_1|> Represent the given image with the following question:  How many people are present in this image?` |
| 🛠 Materials | `<|image_1|> Represent the given image with the following question:  What material are the objects in this image made of?` |
| 🕐 Times | `<|image_1|> Represent the given image with the following question:  What time of day is depicted in this image?` |
| 🌦 Weathers | `<|image_1|> Represent the given image with the following question:  What is the weather like in this image?` |
| 🤸 Gestures | `<|image_1|> Represent the given image with the following question:  What gesture are the people making in this image?` |

Table 11: Human-written prompts for COCO-FACET.

| Categories | Prompts |
|---|---|
| 🕊Animals | `<\|image_1\|> Represent the given image with the following question: What animals are in this image?` |
| 🏖Scenes | `<\|image_1\|> Represent the given image with the following question: What scene is in the image?` |
| 💼Objects | `<\|image_1\|> Represent the given image with the following question: What objects are in the image?` |
| 👥Count of People | `<\|image_1\|> Represent the given image with the following question: How many people are in the image?` |
| 🛠Materials | `<\|image_1\|> Represent the given image with the following question: What are the objects made of in the image?` |
| 🕐Times | `<\|image_1\|> Represent the given image with the following question: When is the image taken?` |
| ⛅Weathers | `<\|image_1\|> Represent the given image with the following question: What is the weather in the image?` |
| 🤸Gestures | `<\|image_1\|> Represent the given image with the following question: What is the person doing in the image?` |

Table 12: Recall@1 and Recall@5 of text-to-image retrieval in percentage points on our COCO-FACET benchmark with no prompt, GPT-written prompts, and human-written prompts. The human-written prompts lead to a similar performance gain.

| | 🕊 | 🏖 | 💼 | 👥 | 🛠 | 🕐 | ⛅ | 🤸 | Avg. |
|---|---|---|---|---|---|---|---|---|---|
| | | | | Recall@1 | | | | | |
| VLM2Vec-Phi-3.5-V | **95.5** | 69.8 | 74.4 | 14.4 | 6.3 | 5.5 | 4.3 | 9.8 | 44.5 |
| w/ GPT prompt | 90.7 | **81.4** | 75.5 | **72.7** | **25.8** | 18.4 | **14.4** | 15.7 | 53.4 |
| w/ human prompt | 93.5 | 80.8 | **75.7** | 68.1 | 24.8 | **43.6** | 14.3 | **22.9** | **56.3** |
| | | | | Recall@5 | | | | | |
| VLM2Vec-Phi-3.5-V | 99.3 | 90.7 | 90.7 | 36.6 | 18.2 | 12.9 | 19.2 | 27.1 | 58.9 |
| w/ GPT prompt | 98.7 | 95.9 | **92.0** | **92.1** | **48.8** | **82.4** | 36.5 | 39.3 | **75.5** |
| w/ human prompt | **99.6** | **95.9** | 91.6 | 91.2 | 43.8 | 64.2 | **36.9** | **50.2** | 74.6 |

## B.3 Promptable Image Embeddings

We list the obtained GPT-written prompts for eight categories of our COCO-FACET benchmark in Table 10. We also test human-written prompts listed in Table 11. The results are shown in Table 12, where we find that human-written prompts can lead to similar improvement.

## B.4 Pre-Processing Embeddings

We use the following template for GPT-4o's prompt selection:

```
{Prompts} Given the instruction {text}, choose the most relevant prompt for
verifying the results. Please answer in one letter.
```

The "Prompts" part lists all the prompts in Table 10 in the format of "A. Represent the given image with the following question: What type of location is depicted in this image?".

Table 13: Recall@1 and Recall@5 of text-to-image retrieval using various base retrievers with promptable image embeddings compared with original image embeddings.

| | 🕊 | ⛱ | 💼 | 👥 | 🛠 | 🕐 | 🌤 | 🏃 | Avg. |
|---|---|---|---|---|---|---|---|---|---|
| | | | | Recall@1 | | | | | |
| VLM2Vec-Phi-3.5-V | 95.5 | 69.8 | 74.4 | 14.4 | 6.3 | 5.5 | 4.3 | 9.8 | 44.5 |
| w/ GPT prompt | 90.7 | 81.4 | 75.5 | 72.7 | 25.8 | 18.4 | 14.4 | 15.7 | 53.4 |
| E5-V | 92.7 | 70.4 | 71.2 | 31.4 | 10.5 | 7.5 | 5.2 | 20.1 | 46.0 |
| w/ GPT prompt | 96.3 | 77.3 | 77.0 | 60.3 | 18.9 | 36.2 | 7.1 | 24.7 | 54.9 |
| MM-Embed | 92.7 | 67.4 | 68.1 | 13.3 | 7.4 | 5.1 | 3.9 | 19.5 | 42.8 |
| w/ GPT prompt | 64.4 | 68.6 | 68.0 | 15.9 | 11.8 | 8.0 | 3.8 | 20.8 | 41.5 |
| MMRet-MLLM-S2 | 97.2 | 72.1 | 76.0 | 29.8 | 10.0 | 8.4 | 3.6 | 24.1 | 48.8 |
| w/ GPT prompt | 91.7 | 78.5 | 82.2 | 78.6 | 27.2 | 21.1 | 8.7 | 23.3 | 57.6 |
| LLaVE-2B | 96.3 | 70.9 | 73.1 | 19.4 | 8.8 | 3.0 | 4.5 | 19.2 | 45.6 |
| w/ GPT prompt | 91.9 | 72.1 | 79.6 | 82.0 | 28.7 | 22.4 | 6.8 | 18.5 | 56.1 |
| | | | | Recall@5 | | | | | |
| VLM2Vec-Phi-3.5-V | 99.3 | 90.7 | 90.7 | 36.6 | 18.2 | 12.9 | 19.2 | 27.1 | 58.9 |
| w/ GPT prompt | 98.7 | 95.9 | 92.0 | 92.1 | 48.8 | 82.4 | 36.5 | 39.3 | 75.5 |
| E5-V | 98.3 | 91.9 | 89.0 | 60.1 | 25.2 | 18.6 | 16.1 | 35.5 | 61.8 |
| w/ GPT prompt | 99.2 | 95.4 | 92.7 | 79.2 | 48.2 | 57.0 | 30.1 | 45.2 | 73.1 |
| MM-Embed | 98.7 | 87.8 | 84.9 | 29.3 | 20.3 | 15.0 | 15.3 | 45.3 | 58.4 |
| w/ GPT prompt | 83.3 | 87.8 | 85.1 | 45.2 | 33.4 | 22.6 | 19.2 | 44.3 | 60.6 |
| MMRet-MLLM-S2 | 99.9 | 92.4 | 91.6 | 45.2 | 28.3 | 19.7 | 17.6 | 49.7 | 64.5 |
| w/ GPT prompt | 98.2 | 95.4 | 96.0 | 94.4 | 52.4 | 75.7 | 33.7 | 45.2 | 77.7 |
| LLaVE-2B | 99.4 | 93.6 | 88.2 | 41.3 | 21.5 | 10.9 | 16.3 | 37.0 | 59.5 |
| w/ GPT prompt | 98.7 | 90.1 | 94.5 | 94.4 | 52.5 | 53.2 | 30.8 | 40.6 | 74.3 |

Table 14: Recall@1 and Recall@5 of FLAIR and CLIP-ViT-B/16 in percentage points on COCO-FACET. FLAIR does not show significant improvement compared with CLIP.

| | 🕊 | ⛱ | 💼 | 👥 | 🛠 | 🕐 | 🌤 | 🏃 | Avg. |
|---|---|---|---|---|---|---|---|---|---|
| | | | | Recall@1 | | | | | |
| CLIP-ViT-B/16 | **94.0** | 44.2 | **57.4** | 1.1 | 1.6 | 3.8 | 3.3 | **9.6** | **35.0** |
| FLAIR | 82.2 | **52.9** | 54.5 | **1.2** | **3.0** | **6.3** | **4.6** | 6.3 | 33.0 |
| | | | | Recall@5 | | | | | |
| CLIP-ViT-B/16 | **99.1** | 75.6 | 77.6 | **14.7** | **10.7** | **17.4** | 12.4 | **20.9** | 49.8 |
| FLAIR | 97.1 | **81.4** | **80.2** | 9.7 | 10.4 | 11.8 | **17.9** | 19.5 | **50.3** |

## C   More results

### C.1   Detailed Results of Various Base Retrievers with Prompt

We show that this strategy generalizes to different base retrievers. See Table 13.

### C.2   Comparison with Other Promptable Embeddings

CLOC [Chen et al., 2024a] and FLAIR [Xiao et al., 2024] are two non-MLLM-based retrievers with similar ideas on promptable image embeddings, but they have constraints on the image focus given prompts: CLOC limits its image focus to be a rectangular region (bounding box)—the prompt is either a bounding box or text, and the text-prompted image embedding is only trained with a single bounding box prediction. This limited image focus is similar to the zooming or cropping approach, which can struggle with non-region-based attributes like ⛱Scenes and 🕐Times. FLAIR relaxes this constraint by training the model with patch-level image-text alignment, but is still restricted to the single-patch alignment, while non-region-based attributes might require a joint match with several patches (e.g., 👥Count of People).

We evaluate FLAIR on COCO-FACET since CLOC is not open-sourced and FLAIR is more flexible with the image focus. We use the same set of GPT prompts as VLM2Vec-Phi-3.5-V. We also

Table 15: Recall@1 and Recall@5 of accelerated text-to-image retrieval with approximated prompt-able image embeddings in percentage points on our COCO-FACET benchmark, along with the standard error. The results of approximation are averaged over five independent runs.

| | 🐦 | ⛵ | 🧳 | 👥 | 🛠 | 🕐 | 🌦 | 🏃 | Avg. |
|---|---|---|---|---|---|---|---|---|---|
| | | | | Recall@1 | | | | | |
| VLM2Vec-Phi-3.5-V | **95.5** | **69.8** | **74.4** | 14.4 | 6.3 | 5.5 | 4.3 | 9.8 | **44.5** |
| w/ linear approx. | 72.1 ± 2.2 | 67.2 ± 0.5 | 57.1 ± 0.4 | **47.5 ± 4.0** | **24.3 ± 1.1** | **35.7 ± 2.7** | **9.0 ± 0.7** | **14.2 ± 2.1** | 42.5 ± 0.5 |
| w/ GPT prompt | 90.7 | 81.4 | 75.5 | 72.7 | 25.8 | 18.4 | 14.4 | 15.7 | 53.4 |
| | | | | Recall@5 | | | | | |
| VLM2Vec-Phi-3.5-V | **99.3** | 90.7 | **90.7** | 36.6 | 18.2 | 12.9 | 19.2 | 27.1 | 58.9 |
| w/ linear approx. | 84.6 ± 2.1 | **91.9 ± 0.6** | 83.2 ± 0.5 | **73.7 ± 1.0** | **43.9 ± 1.8** | **71.5 ± 1.2** | **28.4 ± 0.8** | **38.1 ± 1.3** | **67.0 ± 0.3** |
| w/ GPT prompt | 98.7 | 95.9 | 92.0 | 92.1 | 48.8 | 82.4 | 36.5 | 39.3 | 75.5 |

attach the results for CLIP-ViT-B/16 on the same scale with FLAIR for fair comparison (with the weights at `https://huggingface.co/openai/clip-vit-base-patch16` under MIT License). In Table 14, the improvement brought by FLAIR on top of CLIP is not significant nor consistent on COCO-FACET. This shows the advantage of our pipeline that utilizes MLLM for promptable embeddings with a flexible image focus.

## C.3 Detailed Accelerated Retrieval Results

We list the mean and error bar over five independent runs of the linear approximation in Table 15. The variability mainly comes from the random selection of samples used for deriving the matrix $W$. The error bar is calculated as the standard error (sample standard deviation divided by the square root of number of runs).

We evaluate the actual inference cost of our pipeline on an A6000 GPU. To simulate larger-scale retrieval settings, we duplicate the COCO-FACET image pool multiple times, as the original dataset does not contain enough images. We use linear approximation with $K = 10$ and test both GPU and CPU settings, since one A6000 GPU only supports up to 1M vectors in FAISS. As shown in Table 16 and Table 17, the promptable embedding step ($KF + F$) is the main cost driver when $N$ is small, but retrieval cost becomes more noticeable when $N$ is large (10M and 20M in Table 17). As for the memory cost of the linear approximation, the peak GPU memory usage is 26.5 GB when we derive the promptable embeddings with a batch size of 50. This fits comfortably on a modern 40GB A100 or 48GB RTX 6000. In use cases with tighter resource constraints, the method can be adapted by reducing batch size with larger latency.

Table 16: Average inference time using linearly approximated promptable image embeddings on VLM2Vec-Phi-3.5-V on 🕐Times with varied image pool size. Retrieval is optimized by faiss.IndexFlatIP on an A6000 GPU.

| Image Pool Size | 1K | 1M |
|---|---|---|
| Avg. Per-Query Time (s) | 7.33 | 7.53 |

Table 17: Average inference time using linearly approximated promptable image embeddings on VLM2Vec-Phi-3.5-V on 🕐Times with varied image pool size. Retrieval is optimized by faiss.IndexFlatIP on CPUs.

| Image Pool Size | 1K | 1M | 10M | 20M |
|---|---|---|---|---|
| Avg. Per-Query Time (s) | 7.25 | 7.61 | 11.60 | 36.39 |

## C.4 Visualization Examples

Figure 4 visualizes which image regions the models attend to when matching a query by computing gradients of the image-text similarity score with respect to input pixels. We use VLM2Vec-Phi-3.5-V as an example, which processes images through a multi-crop strategy: one global crop capturing the entire image context, and multiple local crops arranged in a 2×2 spatial grid that capture fine-grained regional details. Each crop is resized to 336×336 pixels and processed independently by the vision encoder. By backpropagating through the similarity score, we obtain gradient magnitudes for each

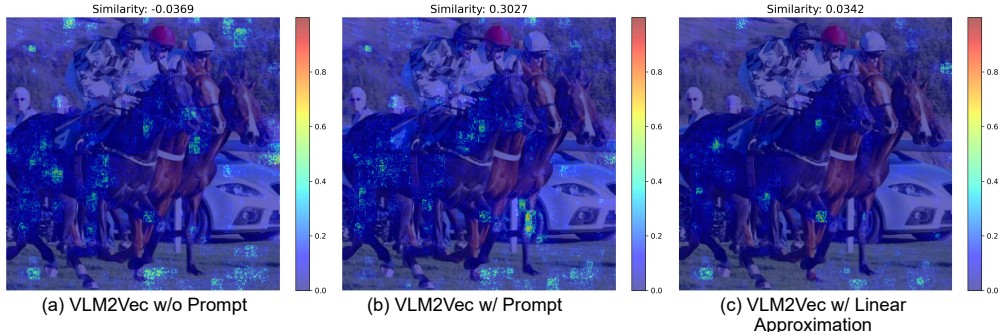

| Similarity: -0.0369 | Similarity: 0.3027 | Similarity: 0.0342 |
| (a) VLM2Vec w/o Prompt | (b) VLM2Vec w/ Prompt | (c) VLM2Vec w/ Linear Approximation |

Figure 4: Visualization of which image regions the models attend to for image-text matching. The query text is "`Find me an image that contains any car.`"

crop, which indicate how sensitive the similarity is to changes in different image regions. The final visualization combines both global and local attention through a weighted average (0.5 for the global map and 0.5 for the local map), producing a comprehensive heatmap that highlights which image regions contribute most to the model's decision.

We observe the change in image focus by comparing the heatmaps in Figure 4. In (a), the image focus is diverse and the image-text similarity is negative. In (b), the prompt shifts the image focus to the front wheel of the car, increasing the similarity to 0.3. In (c), the linear approximation ($K = 100$) is not as effective as using the GPT prompt, but it still reduces the image focus in the irrelevant region (background plants), which leads to a positive similarity.

### C.5 Detailed Text-Based Retrieval Results

We find that text-based retrieval suffers from hallucination and linguistic ambiguity. Some failure cases from `Animals` regarding birds are shown in Figure 5 as examples. We find that elements like sky or feathers could trigger the model to answer with bird's existence. In addition, this approach could not deal with polysemy like `chicken` shown in the fifth example. We also notice that the existence of bird patterns (in the third example) complicates this process, as the model could not prioritize real birds over the bird pattern on the container.

## D Test Cases in COCO-FACET

We collect some challenging test cases in COCO-FACET for better understanding the drawbacks of retrievers. Examples are shown in Figure 6. There are several challenges: (1) The required attributes are not the main subject of the images, especially with the test cases from `Animals` and `Objects`. (2) The semantic understanding of attributes, like "`0 people`" and "`made of.`" (3) The visual grounding of attributes, like "`sports arena`" and "`jumping`". Current retrievers may not ground these attributes precisely. (4) Simple visual reasoning. In the `Count of People` category, the model is required to perform approximation of the number of people in the image. Notably, to reduce the difficulty, we require that the negative samples differ from the positive sample by at least 3, but the model performance is still low. In the example of "`a sunny day,`" the ground truth does not feature a sunny sky but shows clear shadows on the ground. (5) Robustness to deceptive attributes. In the example of "`during the day time,`" there are lights outside in the CLIP-retrieved image, but human can judge that this image was taken at night.

## E Limitations

### E.1 COCO-FACET Benchmark

First, COCO-FACET is built on existing annotations of COCO images. Although we have conducted manual checking detailed in Appendix A, there might still contain inaccurate and ambiguous annotations. For example, some small animals or objects might not be annotated by bounding boxes in

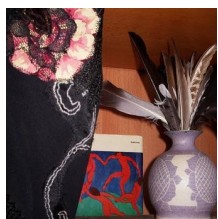

USER: <image>
Which animals can be seen in this image? Please answer in detail. ASSISTANT:
in this image, there are birds visible.

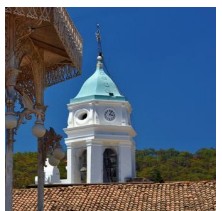

USER: <image>
Which animals can be seen in this image? Please answer in detail. ASSISTANT:
in the image, there are two birds visible. one bird is perched on a light fixture, while the other bird is flying in the sky. the presence of these birds adds a sense of liveliness to the scene, as they interact with the architecture and the surroundings.

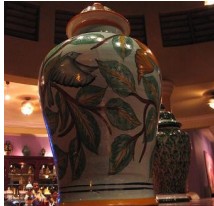

USER: <image>
Which animals can be seen in this image? Please answer in detail. ASSISTANT:
in the image, there are two birds visible. one bird is perched on a branch, while the other bird is sitting on a vase. the birds are part of the decorative elements in the scene, adding a touch of nature to the setting.

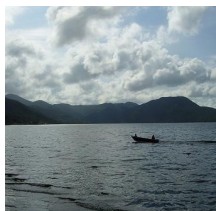

USER: <image>
Which animals can be seen in this image? Please answer in detail. ASSISTANT:
in the image, there are two birds visible. they are flying over the water, adding a sense of movement and life to the scene.

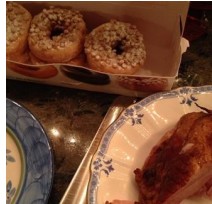

USER: <image>
Which animals can be seen in this image? Please answer in detail. ASSISTANT:
in this image, there is a chicken visible on the plate.

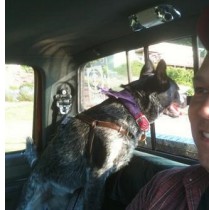

USER: <image>
Which animals can be seen in this image? Please answer in detail. ASSISTANT:
in this image, a dog can be seen.

Figure 5: Failed top-1 retrieval results of the text-based retrieval. The query is "`Find me an image that contains any bird.`" in all cases.

MSCOCO, which could affect the evaluation of the `Animal` and `Object` attributes in COCO-FACET. Second, some existing universal multimodal embedders like mmE5 [Chen et al., 2025] are not evaluated on our benchmark due to the limited computation resource.

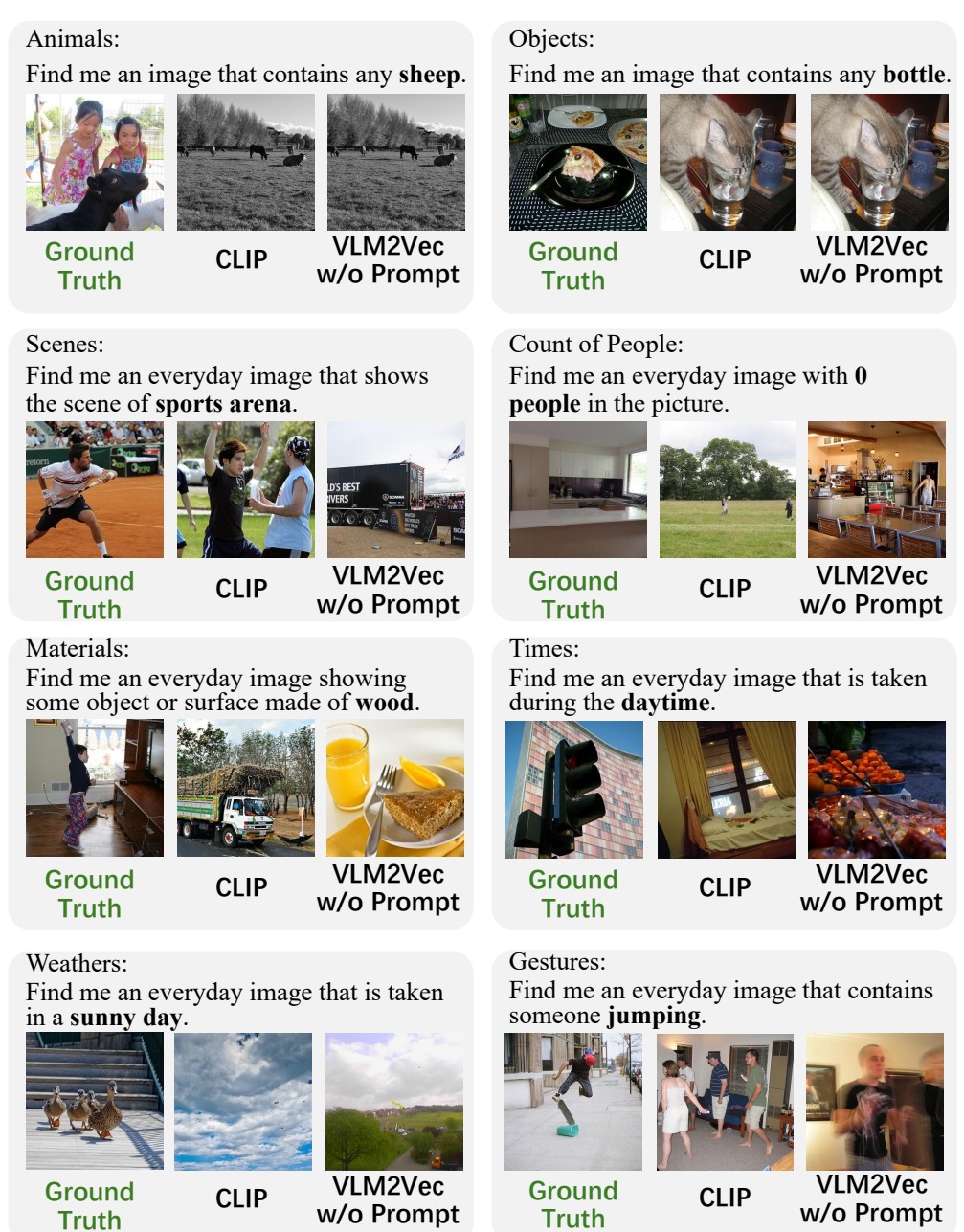

Figure 6: Challenging test cases in COCO-FACET. In each test case, the first image is the ground truth, the second is the top-1 image retrieved by CLIP, and the third is the top-1 image retrieved by VLM2Vec without promptable embeddings. VLM2Vec with GPT prompts can solve these test cases.

## E.2 Promptable Image Embeddings

First, our pipeline in Section 4.1 relies on the usage of GPT-4o's API. While other large language models, especially the open-sourced ones, can be good alternatives, we have not tested them in our scenario yet.

Second, the promptable image embeddings do not fully resolve the imbalance on different attributes, as we observe that the Recall@1 and Recall@5 accuracies for `Material`, `Weather`, and `Gesture` are lower than other attributes.

Table 18: Recall@1 and Recall@5 of selecting pre-processed promptable image embeddings of four disjoint prompts (the last row) and using image embeddings with or without prompt (baseline) in percentage points on COCO-FACET.

| | 🕊 | ⛱ | 🛠 | 🤸 |
|---|---|---|---|---|
| Recall@1 | | | | |
| VLM2Vec-Phi-3.5-V | 95.5 | 69.8 | 6.3 | 9.8 |
| w/ GPT prompt | 90.7 | **81.4** | **25.8** | **15.7** |
| w/ Disjoint Prompt Set | **97.0** | 71.5 | 8.1 | 11.4 |
| Recall@5 | | | | |
| VLM2Vec-Phi-3.5-V | **99.3** | 90.7 | 18.2 | 27.1 |
| w/ GPT prompt | 98.7 | **95.9** | **48.8** | **39.3** |
| w/ Disjoint Prompt Set | **99.3** | 90.1 | 19.5 | 25.4 |

Table 19: Recall@1 and Recall@5 of Linear Approximation on $K = 100$ samples from another random category (the last row), default Linear Approximation (the second row), and without prompts (the first row) in percentage points on COCO-FACET. Results in the last two rows are averaged over five independent runs.

| | 🕊 | ⛱ | 💼 | 👥 | 🛠 | 🕐 | 🌦 | 🤸 | Avg. |
|---|---|---|---|---|---|---|---|---|---|
| Recall@1 | | | | | | | | | |
| VLM2Vec-Phi-3.5-V | **95.5** | **69.8** | **74.4** | 14.4 | 6.3 | 5.5 | 4.3 | 9.8 | **44.5** |
| w/ linear approx. | 72.1 | 67.2 | 57.1 | **47.5** | **24.3** | **35.7** | **9.0** | **14.2** | 42.5 |
| on disjoint samples | 56.9 | 62.4 | 54.8 | 37.7 | 20.4 | 23.2 | 5.4 | 8.4 | 37.4 |
| Recall@5 | | | | | | | | | |
| VLM2Vec-Phi-3.5-V | **99.3** | 90.7 | **90.7** | 36.6 | 18.2 | 12.9 | 19.2 | 27.1 | 58.9 |
| w/ linear approx. | 84.6 | **91.9** | 83.2 | **73.7** | **43.9** | **71.5** | **28.4** | **38.1** | **67.0** |
| on disjoint samples | 75.8 | 89.5 | 83.8 | 63.7 | 41.6 | 60.9 | 22.2 | 27.0 | 61.4 |

Third, the success of accelerated promptable image embeddings is not entirely zero-shot but relies on prior knowledge of the query. For the pre-processing approach, although we show that it is applicable to some unseen attributes, it does limit support for entirely novel and highly specific attributes without prior knowledge. We simulate this setting by randomly splitting the categories on COCO-FACET into two parts, limiting the prompt set to be four random prompts from the original eight and testing on the other four categories. This leads to a performance drop compared to the vanilla approach with ground-truth prompts (Table 18). For the linear approximation approach, $W$ is derived from images of the same category as the query. We test the case when the categories of available samples are disjoint from the inference categories by randomly choosing another category and then uniformly sampling $K = 100$ samples from its pool. In Table 19, we observed that using such samples does not render the method completely ineffective, and still outperforms the baseline (without prompts, the first row) at Recall@5. However, there is a decline compared with using images from the same category. Hence, when there is no image from the target category, the linear approximation does not have a guaranteed performance.

# F    Broader Impacts

Improving attribute-focused text-to-image retrieval can benefit applications that rely on fine-grained visual understanding, such as e-commerce. Our method enhances the precision of such retrieval tasks while maintaining efficiency, potentially enabling more responsive and accurate systems.

At the same time, fine-grained retrieval poses risks, including potential misuse in surveillance or amplification of biases. In addition, since our approach builds on pretrained multimodal models like Phi-3.5-V, it may inherit existing biases and vulnerability to adversarial attacks of such models.

To support responsible use, we encourage transparency around deployment contexts and recommend auditing tools to monitor for unintended outcomes. We release our benchmark and code to facilitate further research on both the benefits and limitations of attribute-focused retrieval.

