# OpenReview forum: "Highlighting What Matters: Promptable Embeddings for Attribute-Focused Image Retrieval"
_NeurIPS.cc/2025/Conference — NeurIPS 2025 poster_

### Official Review · Reviewer_roin · 2025-06-28

**Clarity:** 2
**Significance:** 2
**Originality:** 2
**Rating:** 4
**Confidence:** 4

**Summary:**

This paper introduces promptable image embeddings to enhance attribute-focused text-to-image (T2I) retrieval. The authors propose a new benchmark, COCO-FACET, and show that existing CLIP-like and multimodal retrieval models perform suboptimally on fine-grained attribute queries. They demonstrate that conditioning image embeddings on attribute-specific prompts substantially improves retrieval accuracy. Additionally, two acceleration strategies are presented to facilitate practical deployment.

**Questions:**

Please refer to the section outlining the weaknesses of the paper. I look forward to the authors' responses to these questions.

**Ethical Concerns:**

["NO or VERY MINOR ethics concerns only"]

**Final Justification:**

The response addresses most of my concerns. Hence, I have raised my score to 4.

**Limitations:**

yes

**Quality:**

3

**Strengths And Weaknesses:**

Pros:
1. The authors clearly identify the limitations of global image embeddings and propose a well-motivated approach, which is highly relevant for T2I retrieval tasks.
2. The introduction of the COCO-FACET benchmark is a valuable contribution, as it encourages further research into the retrieval of fine-grained visual attributes.

Cons:
1. The proposed method is somewhat similar to previous works, such as CLOC and FLAIR, which also generate task-specific embeddings. The authors should more explicitly clarify the distinctions and advantages of their approach over existing methods.
2. The method utilizes GPT-4o to generate prompts that guide image embeddings. However, since GPT-4o is itself a state-of-the-art multimodal large language model, there is a concern that GPT-4o alone might already solve the retrieval task independently, without the need for additional models like VLM2Vec. The paper does not evaluate this possibility or clarify whether GPT-4o serves solely as a prompt generator or is effectively part of the retrieval pipeline.
3. The inclusion of acceleration strategies is a positive aspect. However, the paper lacks a detailed comparison of the actual inference time between the proposed approach and prior methods. Providing such a comparison would enhance the practical significance of the work.
4. While the authors state that their method can handle attributes such as weather and time, supported by experimental results, the paper does not clearly explain how promptable embeddings capture these global or abstract concepts. A more detailed explanation or visualization of how these attributes are encoded, as well as an analysis of failure cases, would strengthen this point.

---

> ### Author Rebuttal · Authors · 2025-07-31
>
> Thank you for taking the time to review our paper, and we appreciate your feedback and suggestions! First of all, we would like to clarify a possible misunderstanding in your review:
>
> - *"However, since GPT-4o is itself a state-of-the-art multimodal large language model, there is a concern that GPT-4o alone might already solve the retrieval task independently, without the need for additional models like VLM2Vec."*
>
>     GPT-4o is used only as the prompt generator (and prompt selector in the pre-processing approach) without access to any image in our pipeline. It can be replaced by pure-language models with instruction-following ability. Hence, there is no such possibility that it is the actual solver of the retrieval task, instead of the promptable retriever.
>
> Then we address your concerns as follows:
>
> 1. **Your concern on the distinctions and advantages of our approach over existing methods (CLOC and FLAIR).**
>
>     Both CLOC [1] and FLAIR [2] have constraints on the image focus given prompts. CLOC proposed the idea of promptable image embeddings, but limited its image focus to be a rectangular region (bounding box). As introduced in their Section 3.2, their prompt is either a bounding box or text, and the text-prompted image embedding is only trained with a single bounding box prediction. This limited image focus is similar to the zooming or cropping approach, which would not work well on non-region-based attributes like Time and Scene, as we mentioned in lines 108-111. FLAIR relaxed this constraint by patch-level image-text alignment during training but is restricted to single-patch alignment, while non-region-based attributes might require a joint match with several patches (e.g., Count of People).
>
>     To illustrate the distinction, we evaluated FLAIR on our COCO-Facet benchmark (since CLOC is not open-sourced and FLAIR is more flexible with the image focus due to its objective). As the promptable retrievers in the paper are not on the same scale, we evaluated the CLIP-ViT-B/16 model on the same scale with FLAIR for fair comparison. We attached the results in Table 1 and Table 2. The improvement brought by FLAIR on top of CLIP is not significant or consistent on the COCO-Facet benchmark, and overall it still has poor performance.
>
>     *Table 1: Recall@1 of FLAIR and CLIP-ViT-B/16 (baseline) in percentage points on COCO-Facet.*
>     | |Animals | Scenes | Objects | Count of People | Materials | Times | Weathers | Gestures | Avg.|
>     | ---- |:----:|:-----:|:-----:|:-----:|:-----:|:-----:|:-----:|:-----:|:-----:|
>     | CLIP-ViT-B/16 | 94.0 |  44.2 |  57.4  | 1.1  |  1.6  | 3.8 | 3.3 |  9.6 | 35.0|
>     | FLAIR | 82.2 | 52.9 | 54.5  |  1.2  |  3.0  | 6.3  | 4.6 | 6.3 | 33.0|
>
>     *Table 2: Recall@5 of FLAIR and CLIP-ViT-B/16 (baseline) in percentage points on COCO-Facet.*
>     | |Animals | Scenes | Objects | Count of People | Materials | Times | Weathers | Gestures | Avg.|
>     | ---- |:----:|:-----:|:-----:|:-----:|:-----:|:-----:|:-----:|:-----:|:-----:|
>     | CLIP-ViT-B/16 | 99.1| 75.6  |  77.6  | 14.7 |  10.7  | 17.4 |  12.4  | 20.9 |49.8 |
>     | FLAIR | 97.1 | 81.4 |  80.2  |  9.7  |  10.4  |11.8  | 17.9  | 19.5 |50.3 |
>
> 2. **Your concern on the possibility that GPT-4o might already solve the task.**
>
>     As we clarified above, the GPT-4o only functions as the prompt generator (and prompt selector in the pre-processing approach) without looking at any image, and we attach the prompts it generated in Table 10 for your reference. Therefore, the improvement is brought by the promptable image embeddings derived by the retrievers instead of the multimodal LLM as the prompt generator.
>
> 3. **Your concern on the actual inference time between the proposed approach and prior methods.**
>
>     We thank the reviewer for the suggestion of reporting the actual inference time among the methods. We averaged the inference time over queries from the Times category on a single A6000 GPU. Note that COCO-Facet has a small image pool size (N=100) per query for fast evaluation, and thus K=100 resulted in the same processing time. So we added the setting of K=10 to reflect the difference:
>
>     *Table 3: Average inference time of VLM2Vec-Phi-3.5-V without prompt, with prompt, and with linear approximation (K=10).*
>     |  | Avg. Per-Query Time|
>     | ---- |:---:|
>     | w/o Prompt |0.04s |
>     | w/ Prompt | 17.2s |
>     | w/ Linear Approximation (K=10) | 6.8s |
>
>     We can see that the Linear Approximation allows acceleration when $K<N$. We will include the table in the Appendix.
>
> 4. **Your question on the explanation or visualization of how these attributes are encoded, as well as an analysis of failure cases.**
>
>     Since the MLLM-based universal embedders are previously shown to be capable of encoding question/instruction+image for composed image retrieval (e.g., CIRR [3], the query text being "Show three bottles of soft drink", the query image being one bottle as an example, and the target image being three bottles in the same style), they integrate the textual and visual information in a similar way that the original MLLM did in QA tasks, instead of doing direct addition. Hence, when we use attribute-focused questions as text inputs, we expect that the output embedding would be attribute-focused image embeddings.
>
>     We agree that a visualization would help the reader understand the promptable approach better. We have attached some case analysis of COCO-Facet in Figure 4 of Appendix D. Following your suggestion, we will include some failure case analysis in the Appendix to better demonstrate the scope of our approach.
>
> If you have further questions or concerns, we would be happy to address them!
>
> [1] Chen et al., Contrastive Localized Language-Image Pre-Training, arXiv 2024.
>
> [2] Xiao et al., FLAIR: VLM with Fine-grained Language-informed Image Representations, CVPR 2025.
>
> [3] Liu et al., Image Retrieval on Real-life Images with Pre-trained Vision-and-Language Models, ICCV 2021.

---

> > ### Comment · Reviewer_roin · 2025-08-02
> >
> > The response addresses most of my concerns. Hence, I have raised my score to 4. Good luck!

---

> > > ### Author Response · Authors · 2025-08-02
> > >
> > > Thank you for your feedback and for reconsidering your score! We are glad to know that we addressed most of your concerns. If there are any remaining issues or suggestions, we are happy to hear them as well.

---

### Official Review · Reviewer_Lakj · 2025-06-30

**Clarity:** 3
**Significance:** 2
**Originality:** 3
**Rating:** 4
**Confidence:** 4

**Summary:**

This paper focuses on the attribute-focused T2I retrieval task, where queries emphasize specific attributes of the target image. Motivated by this setup, the authors introduce the COCO-FACET benchmark, built upon the COCO dataset, to evaluate performance in this setting. They also propose a promptable embedding method with MLLM model. I believe the motivation is strong and dataset can be a contribution, but there are some concerns on assumptions that used in method

**Questions:**

I wrote them above

**Ethical Concerns:**

["NO or VERY MINOR ethics concerns only"]

**Final Justification:**

I carefully read all the responses and reviews. My main concern was the computational overhead that this method possibly has, but the rebuttal has largely resolved my concern on this, so I will raise my point.

**Limitations:**

I wrote them above

**Paper Formatting Concerns:**

I wrote them above

**Quality:**

3

**Strengths And Weaknesses:**

**Strengths**
1. The targeted problem (attribute-focused) T2I task seems important and reasonable.
2. The suggested dataset seems reasonable and constructed carefully to cope with intended goal.
3. The paper is easy to follow and the related works are clearly explained.
4. The baselines are fairly well compared with the recent ones.


**Weaknesses**
1. **Scalability and Assumptions of the Method**
A key concern lies in the computational assumptions underlying the proposed method. As the authors themselves note, the promptable embeddings need to be recomputed for the entire gallery set for each query—if this is indeed how the original implementation works, it would be infeasible for large-scale gallery sets common in real-world applications. Even with the proposed linear approximation, the computational cost remains significantly higher than traditional T2I retrieval methods—potentially up to 100× more expensive for K=100.
Furthermore, both acceleration strategies rely on assumptions that may not hold in practice, such as prior knowledge of the target category or the ability to pre-filter gallery images by category. These assumptions undermine the method’s applicability in realistic settings. The paper would benefit from an alternative approach that avoids recomputing gallery representations for each query. I think the current direction for the method would not be feasible in even a moderate size index set (gallery) setting.

2. **Dataset Visualization and figure**
More concrete visualizations of the dataset would improve clarity and help readers better understand the proposed benchmark. Additionally, Figure 1 is potentially misleading, as the query text appears to be embedded within the gallery set, which may confuse readers about the retrieval setup.

---

> ### Author Rebuttal · Authors · 2025-07-31
>
> Thank you for taking the time to review our paper, and we appreciate your feedback and suggestions! First of all, we would like to clarify a possible misunderstanding in your review regarding our approach:
>
> - *"Even with the proposed linear approximation, the computational cost remains significantly higher than traditional T2I retrieval methods—potentially up to 100× more expensive for K=100 ... The paper would benefit from an alternative approach that avoids recomputing gallery representations for each query. I think the current direction for the method would not be feasible in even a moderate size index set (gallery) setting."*
>
>    The linear approximation is not 100× more expensive than prior methods for K=100 in terms of total per-query cost. For the approximation, we only need to recalculate the promptable embeddings of K images randomly selected from the pool, so the per-query embedding cost is $KF + F$ (lines 305-306), which **does not grow with the index set size $N$**. Besides the embedding cost, we need to perform cosine similarity search over the entire index set, which costs $N \times D$ for $D$-dim embeddings in the exact search setting (FAISS's IndexFlatIP). Hence, the per-query embedding cost is not always the bottleneck in the total per-query cost. Mathematically, the new total per-query cost is $KF+F+ND$ versus prior $F+ND$, which is not 100x more expensive, especially when the index set size $N$ is very large (a common case in a realistic setting).
>
> Then we address your concerns and respond to your suggestions as follows.
>
> 1. **Your concern about the assumptions of the two acceleration strategies that may not hold in practice.**
>
>     For the pre-processing approach, we do not assume that the target category is known exactly, since we include GPT-4o for prompt selection at test time with the template attached in Appendix B.4. On the other hand, we agree that pre-defined attributes might not work for entirely novel and highly specific attributes disjoint from the prompt set, but still they show applicability to unseen but similar attributes. In lines 286-288 and Table 5, we observe that using the exact ground-truth prompt is often not necessary, as "Count of People" prompt works well on many test cases in "Gesture".
>
>     As a fallback strategy for novel attributes, the on-the-fly linear approximation approach samples the set of K images uniformly from the candidate pool across all queries in the category (much larger than the candidate pool of one query). But it is not necessary to sample the images from the same category (or "pre-filter gallery images by category" as you mentioned) to achieve better performance on COCO-Facet. To illustrate this point, we show the results of sampling K images from a randomly chosen category other than the target one. In Table 1 and 2, we found that this does not lead to complete failure, and outperforms the baseline at Recall@5.
>
>     *Table 1: Recall@1 of Linear Approximation on K=100 samples from another random category (disjoint), default Linear Approximation, and without prompts on VLM2Vec-Phi-3.5-V in percentage points on COCO-Facet. Results in the last two rows are averaged over five independent runs.*
>     | |Animals | Scenes | Objects | Count of People | Materials | Times | Weathers | Gestures | Avg.|
>     | ---- |:----:|:-----:|:-----:|:-----:|:-----:|:-----:|:-----:|:-----:|:-----:|
>     | No Prompt | **95.5** | **69.8**  |  **74.4** | 14.4 | 6.3  | 5.5 | 4.3 | 9.8  | **44.5**|
>     | Default |  72.1 | 67.2  | 57.1  |**47.5**  | **24.3**  | **35.7** | **9.0** | **14.2**  | 42.5|
>     | Disjoint | 56.9 | 62.4   | 54.8 |37.7 |  20.4 |23.2 | 5.4 | 8.4 | 37.4 |
>
>     *Table 2: Recall@5 of Linear Approximation on K=100 samples from another random category (disjoint), default Linear Approximation, and without prompts on VLM2Vec-Phi-3.5-V in percentage points on COCO-Facet. Results in the last two rows are averaged over five independent runs.*
>     | |Animals | Scenes | Objects | Count of People | Materials | Times | Weathers | Gestures | Avg.|
>     | ---- |:---:|:----:|:----:|:----:|:----:|:---:|:---:|:---:|:---:|
>     | No Prompt | **99.3** | 90.7  | **90.7**  | 36.6 | 18.2  |12.9  |19.2  | 27.1  |58.9 |
>     | Default | 84.6 | **91.9**  | 83.2  |**73.7**  | **43.9**  | **71.5** | **28.4** | **38.1**  |**67.0** |
>     | Disjoint  | 75.8 | 89.5 | 83.8  |63.7  |  41.6 | 60.9 |22.2 | 27.0 | 61.4 |
>
> 2. **Your suggestion on the clarity of Figure 1.**
>
>     Thank you for your suggestion! For visualization of the benchmark, we included some examples from the benchmark and failure analysis in Figure 4 of Appendix D. For Figure 1, we will replace the bottom-left text "Image" with multiple images representing the gallery set to help readers better understand our approach.
>
> If you have further questions or concerns, we would be happy to address them!

---

> ### Comment · Reviewer_Lakj · 2025-08-03
>
> Thank you for the thorough rebuttal.
> However, I still have concerns regarding the computational overhead of the proposed method.
>
> 1. In the first response, I meant that the overhead depends on K (not N) in the case of linear approximation. But I also wanted to highlight that, even in this case, as the authors noted, it depends on KF relative to the original computation F. Specifically, as the authors point out, there is an inherent comparison between KF+F+ND vs. F+ND, and it would not be perfectly multiplied for K. But, to concretely claim this, I believe the authors should provide experimental results for when N becomes large. Additionally, the relative percentages of KF and ND may change depending on the dataset and retrieval setting. In the reasonable (practical) size database, I believe KF would be the main factor given that ND can be optimized using FAISS library.
> For instance, in the current dataset (COCO-FACET), as the authors mentioned in the response to reviewer roin’s comments, even with the linear approximation, the query time varies significantly (0.04s vs. 6.8s), which indicates a considerable computational overhead, which makes me concerned. Therefore, I believe a more detailed explanation of the computational overhead analyses (including memory overhead, which is omitted now) is necessary to address my concerns.
>
> --> I generally appreciate the problem setting presented here, but I am uncertain whether this time-consuming proposed method is necessary to solve the problem and it is practically acceptable.
>
> 2. I think I need more concrete explanation on what the differences are between "Default" and  "Disjoint" here?...

---

> > ### Author Response · Authors · 2025-08-05
> >
> > Thank you for your reply! We address your concerns and questions as follows:
> >
> > 1. **Your concern about the computational overhead and the necessity of our method.**
> >
> >     Below, we clarify the computational overhead by presenting additional analysis. First, we provide the experimental results for **when $N$ becomes large** and use the Times category as an example. To simulate larger-scale retrieval settings, we duplicate the COCO-Facet image pool multiple times, as the original dataset does not contain that many unique images. We use linear approximation with $K=10$ and test both GPU and CPU settings, since our A6000 GPU only supports up to 1M vectors in FAISS. As shown in Tables 1 and 2, the promptable embedding step ($KF + F$) is the main cost driver when $N$ is small, but retrieval cost becomes more noticeable when $N$ is large (10M and 20M in Table 2).
> >
> >     *Table 1: Average inference time using linearly approximated promptable image embeddings on VLM2Vec-Phi-3.5-V on the Times category of COCO-Facet with varied image pool size. Retrieval is optimized by faiss.IndexFlatIP on the same single A6000 GPU.*
> >     | Image Pool Size  | 1K  | 1M |
> >     | ---- |:----:|:-----:|
> >     | Avg. Per-Query Time  | 7.33s |  7.53s  |
> >
> >     *Table 2: Average inference time using linearly approximated promptable image embeddings on VLM2Vec-Phi-3.5-V on the Times category of COCO-Facet with varied image pool size. Retrieval is optimized by faiss.IndexFlatIP on CPUs.*
> >     | Image Pool Size  | 1K  | 1M | 10M | 20M |
> >     | ---- |:----:|:-----:|:-----:|:-----:|
> >     | Avg. Per-Query Time  | 7.25s | 7.61s | 11.60s | 36.39s |
> >
> >     Second, as noted by the reviewer, the relative cost of promptable embedding ($KF$) and retrieval ($ND$) depends on the retrieval setup. To demonstrate **the trade-off between cost and performance**, we vary the sample size $K$ used in the linear approximation. As shown in Table 3, even with small $K$ values, the method improves over the $K=0$ baseline (no prompts). The performance continues to increase with larger $K$, allowing tuning $K$ based on available compute and desired latency. These results support that the method is applicable under varying resource constraints.
> >
> >     *Table 3: Recall@1 and Recall@5 of the accelerated text-to-image retrieval using approximated promptable image embeddings on VLM2Vec-Phi-3.5-V with different sample size K on the Times category of COCO-Facet. $K=0$ corresponds to no prompt. Results are averaged over five independent runs.*
> >     | K |0 | 5 | 10 | 20 | 40 | 60 | 80 | 100 |
> >     | ---- |:----:|:-----:|:-----:|:-----:|:-----:|:-----:|:-----:|:-----:|
> >     | Recall@1 | 5.5 | 3.8 |  22.6 | 15.3 | 24.8 | 26.2 | 27.9 | 35.7|
> >     | Recall@5 | 12.9 | 17.2 | 38.9 | 49.4 | 60.6 | 61.6 | 59.6 | 71.5|
> >
> >     As for the **memory cost** of the linear approximation, we measured the GPU memory usage during query time (including both the approximation and the retrieval) using $K=100$ and the original image pool in the Times category. When deriving the promptable embeddings, we used a batch size of 50. The peak GPU memory usage is **26.5 GB**, which fits comfortably on a modern 40GB A100 or 48GB RTX 6000. In use cases with tighter resource constraints, the method can be adapted by reducing batch size with larger latency.
> >
> >     From the performance side, we have shown that attribute-focused queries are challenging for CLIP-like retrievers and MLLM-based retrievers without prompt. We further strengthened this point by presenting the results of FLAIR, a retriever designed for highlighting the important part by task-specific embeddings, which offered little improvement. Our method as an attempt to solve this task significantly improves the performance, which justifies the additional cost. Moreover, the linear approximation enables cost-flexible deployment, offering a practical trade-off between quality and latency.
> >
> > 2. **Your question on the experiment setting in our rebuttal.**
> >
> >     The "Disjoint" setting means applying linear approximation on $K=100$ random samples from another random category (uniformly chosen from the remaining seven categories) rather than the same category with the query. The "Default" setting is the one we used in the paper, which applies approximation on $K=100$ random samples from the same category with the query. The purpose of this experiment is to clarify that for the linear approximation approach, using samples from the target category (or the assumption you mentioned as "the ability to pre-filter gallery images by category") is not always necessary for a better performance compared with the baseline.
> >
> > If you have further questions or concerns, we would be happy to address them as well!

---

> > > ### Comment · Reviewer_Lakj · 2025-08-07
> > >
> > > Thanks for the detailed response. My main concerns are largely resolved, so I will raise my score to 4. Please include the concrete computational analyses in the revised version.

---

> > > > ### Author Response · Authors · 2025-08-07
> > > >
> > > > Thank you for your follow-up and for reconsidering your score! We are glad to hear that your main concerns have been addressed. As suggested, we will include the computational analyses in the revised version.

---

### Official Review · Reviewer_mpkJ · 2025-07-02

**Clarity:** 2
**Significance:** 3
**Originality:** 2
**Rating:** 4
**Confidence:** 4

**Summary:**

The paper presents a new benchmark called COCO-Facet which focuses on fine-grained attributes for text-image retrieval tasks. The retrieval task aims at difficult categories such as gestures, times, material, and counts, among others, revealing that current VLM embedding models struggle with identifying these attributes reliably. Finally, the paper proposes to use MLLM embedding models with promptable embeddings to improve the retrieval performance significantly and offers insights about accelerating the inference speed of these large models.

**Questions:**

1. In Sec. 5.1, how does the performance change if you do not assume knowledge about the categories, i.e., the prompt set is disjoint from the GT categories.
2. How do you choose the (a,b) pairs for Sec. 5.2? What happens if the categories for which K samples are available is disjoint from the inference categories, i.e., novel tasks/categories at test time?
3. From the cited related work, it seems that FLAIR [2] is the closest to a promptable embedding model that is not based on an MLLM. Since it is concurrent work, I do not view it negatively that it is missing from the comparison, but it would certainly be interesting to see how it compares in Tab. 1 and/or Tab. 2. However, it does not seem to be available in a comparable model size and is a smaller model.

[2] Xiao et al., FLAIR: VLM with Fine-grained Language-informed Image Representations, CVPR 2025

**Ethical Concerns:**

["NO or VERY MINOR ethics concerns only"]

**Final Justification:**

The rebuttal revealed some valuable insights about the importance of prior knowledge for the proposed prompting strategy.
In the "disjoint" setting the performance gains are not nearly as strong and sometimes even weaker than without a prompt.
As a results, the proposed method is not as strong as the paper might make it appear to be.

I believe it is important to discuss these findings as part of a limitation of the proposed method and direction for future work to close the gap between the zero-shot setting and with prior knowledge.

If this important findings is not transparently communicated and disclosed in the paper, I don't think it should be published, so I maintain my initially positive rating under the expectation that the authors incorporate this findings into the paper.

**Limitations:**

Yes

**Quality:**

3

**Strengths And Weaknesses:**

### Strengths
- The proposed COCO-Facet dataset is well motivated, tackles an important problem of fine-grained attribute retrieval, and facilitates further development of stronger embedding models.
- The quantitative evaluation is extensive, covering a wide range of VLMs, both native embedding models and MLLM adapted embedding models.
- Employing promptable models is a reasonable step towards solving this task. The paper also addresses the inference speed issue by providing two possible solutions with their own trade-offs: pre-processing embeddings and linear approximation.

### Weaknesses
- **Missing comparison to existing benchmarks**
  - The paper does not justify the introduction of this new dataset with respect to existing ones. For instance, MMVP [1] contains similar categories such as counts, color/appearance, structural characteristics, etc. A discussion around the new aspects this dataset introduces compared to existing benchmarks is important for judging its value.
  - Comparing the proposed method improvements on related datasets could further reveal if the behavior of the improvements is similar or whether one dataset is more challenging than the other.
- **Methods should be evaluated zero-shot**
  - The proposed method make a lot of assumptions about prior knowledge about the inference task which is often not the case in real-world scenarios. For instance, the pre-processed embeddings contain the GT best-case prompt as part of the prompt set simplifying the problem significantly. Moreover, it seems the category of the retrieval query needs to be known at test time for the promptable embedding solutions.
  - A more realistic setting would either test the pre-processed embeddings from COCO-Facet on another dataset, or the other way around, possibly even creating a uninformed prompt set independently.
- **Technical contributions are limited**
  - Since the main contribution is the dataset, I see this as a minor weakness. Nonetheless, promptable embedding are a key feature of the MLLM embedding models, although Sec. 4 makes it sounds like something innovative.
  - Sec. 5.2 is an interesting approach, but lacks in experimental details and clarity. It seems the approach requires a base dataset of K samples per category but it is not explained how this set is chosen and how one would know the category of the inference sample. Related to the previous point, in realistic applications, training and test set might not come from the same distribution.

[1] Tong et al., Eyes Wide Shut? Exploring the Visual Shortcomings of Multimodal LLMs, CVPR 2024

---

> ### Author Rebuttal · Authors · 2025-07-31
>
> Thank you for taking the time to review our paper, and we appreciate your feedback and suggestions! We address your concerns as follows:
>
> 1. **Your concern on the comparison of COCO-Facet to existing benchmarks.**
>
>     First, compared with commonly used text-to-image retrieval benchmarks (e.g., COCO and Flickr30K), COCO-Facet contains queries on uncommon or non-dominant attributes, which is our motivation for constructing it (described in the beginning of Section 3, lines 113-117).
>
>     Second, compared with MMVP [1], COCO-Facet differs in motivation, task definition, and scale. COCO-Facet evaluates how well current models handle attribute-focused **text-to-image retrieval** queries, with 100 candidate images for each (random guess would get 1\%). MMVP focuses on fine-grained **image-text matching**, where there are only two images and two queries per testcase for evaluating "whether the VLM can distinguish two visually similar images" (random guess would get 25\%). Due to this difference, the model performance on them cannot be translated into each other directly. Moreover, COCO-Facet is on a scale 30x larger (9,112 queries to 270 questions in MMVP-VLM) which is thus less sensitive to noise or random fluctuations.
>
>     Nonetheless, we show that the improvement of the promptable approach generalizes to the image-text matching task in MMVP benchmark. We use the given questions of MMVP in the prompt for the promptable approach. The average performance of the promptable approach is 44.4, which is higher than the best result reported in their paper (39.3 by DFN ViT-H-14). However, due to the different nature of MMVP and COCO-Facet, we cannot draw the conclusion about their relative difficulty based on the relative improvement.
>
>     *Table 1: Results of CLIP-ViT-L/14-336px (for comparison), VLM2Vec-Phi-3.5-V with and without prompt on MMVP-VLM.*
>
>     |  | Orientation & Direction |Presence of Specific Features |State & Condition |Quantity & Count| Positional & Relational Context | Color & Appearance |Structural & Physical Characteristics |Texts | Viewpoint & Perspective| MMVP Avg. |
>     | --- |:--:|:--:|:--:|:--:|:--:|:--:|:--:|:--:|:--:|:--:|
>     | CLIP-ViT-L/14-336px | 0.0 | 20.0 | 40.0 | 20.0 | 6.7 | 20.0 | 33.3 | 6.7 | **33.3** | 20.0|
>     | VLM2Vec-Phi-3.5-V |6.7 |13.3|26.7|33.3|6.7|**60.0**| 20.0|26.7 |6.7|22.2  |
>     | w/ MMVP Question as Prompt  | **26.7** | **46.7** |**60.0**|**73.3** |**26.7** |**60.0** |**40.0** |**40.0**| 26.7|**44.4** |
>
> 2. **Your concern on assumptions about prior knowledge about the inference task.**
>
>     We agree that pre-defined attributes do limit support for entirely novel and highly specific attributes, but still they show applicability to unseen but similar attributes. In lines 286-288 and Table 5, we observe that choosing the exact ground-truth prompt is often not necessary, as "Count of People" prompt works well on many test cases in "Gesture".
>
>     We test the experimental design you proposed: we reevaluate the pre-processing strategy on COCO-Facet, but randomly split the categories into two parts, limiting the prompt set to be four random prompts from the original eight and testing on the other four categories. This setting leads to a performance drop compared to the vanilla approach with ground-truth prompts, but its Recall@1 exceeds the baseline, and the Recall@5 is on par with the baseline. Thus, for cases where the prompt is highly novel and only known at test time, we recommend using the linear approximation as a fallback option.
>
>     *Table 2: Recall@1 of selecting pre-processed promptable image embeddings of four prompts and using image embeddings with or without prompt (baseline) on VLM2Vec-Phi-3.5-V in percentage points on COCO-Facet.*
>     | |Animals | Scenes | Materials | Gestures |
>     | ---- |:-----:|:-----:|:-----:|:-----:|
>     | No Prompt | 95.5 | 69.8 | 6.3  | 9.8  |
>     | w/ GT Prompt | 90.7 | **81.4** | **25.8**  | **15.7**  |
>     | Disjoint Prompt Set | **97.0** |  71.5 |  8.1  | 11.4  |
>
>     *Table 3: Recall@5 of selecting pre-processed promptable image embeddings of four prompts and using image embeddings with or without prompt (baseline) on VLM2Vec-Phi-3.5-V in percentage points on COCO-Facet.*
>     | |Animals | Scenes  | Materials| Gestures |
>     | ---- |:---:|:----:|:---:|:---:|
>     | No Prompt | **99.3**| 90.7   | 18.2   | 27.1  |
>     | w/ GT Prompt | 98.7 | **95.9** | **48.8**  | **39.3**  |
>     | Disjoint Prompt Set  | **99.3** |90.1 | **19.5** | 25.4|
>
>
>
> 3. **Your concern on the technical contributions of the promptable embedding approach, and details about the Linear Approximation.**
>
>     Although MLLM-based embedders were designed and used by many previous papers, with some trying hand-written, domain-specific (news and fashion) prompts and reporting improvements, we are the first to systematically study this approach for attribute-focused embedding of target images using a pipeline of GPT-generated prompts, demonstrating its effectiveness even for some uncommon and challenging attributes.
>
>     For the Linear Approximation approach, we sampled the set of K images uniformly from the pool of candidates of all queries in the category (much larger than the candidate pool of one query). We test the case when the categories of available samples are disjoint from the inference categories by randomly choosing another category and then uniformly sampling K=100 samples from its pool. In Table 2 and 3, we observed that using samples disjoint from the inference category does not render the method completely ineffective, and still outperforms the baseline at Recall@5.
>
>     *Table 4: Recall@1 of Linear Approximation on K=100 samples from another random category (disjoint), default Linear Approximation, and without prompts on VLM2Vec-Phi-3.5-V in percentage points on COCO-Facet. Results in the last two rows are averaged over five independent runs.*
>     | |Animals | Scenes | Objects | Count of People | Materials | Times | Weathers | Gestures | Avg.|
>     | ---- |:----:|:-----:|:-----:|:-----:|:-----:|:-----:|:-----:|:-----:|:-----:|
>     | No Prompt | **95.5** | **69.8**  |  **74.4** | 14.4 | 6.3  | 5.5 | 4.3 | 9.8  | **44.5**|
>     | Default |  72.1 | 67.2  | 57.1  |**47.5**  | **24.3**  | **35.7** | **9.0** | **14.2**  | 42.5|
>     | Disjoint | 56.9 | 62.4   | 54.8 |37.7 |  20.4 |23.2 | 5.4 | 8.4 | 37.4 |
>
>     *Table 5: Recall@5 of Linear Approximation on K=100 samples from another random category (disjoint), default Linear Approximation, and without prompts on VLM2Vec-Phi-3.5-V in percentage points on COCO-Facet. Results in the last two rows are averaged over five independent runs.*
>     | |Animals | Scenes | Objects | Count of People | Materials | Times | Weathers | Gestures | Avg.|
>     | ---- |:---:|:----:|:----:|:----:|:----:|:---:|:---:|:---:|:---:|
>     | No Prompt | **99.3** | 90.7  | **90.7**  | 36.6 | 18.2  |12.9  |19.2  | 27.1  |58.9 |
>     | Default | 84.6 | **91.9**  | 83.2  |**73.7**  | **43.9**  | **71.5** | **28.4** | **38.1**  |**67.0** |
>     | Disjoint  | 75.8 | 89.5 | 83.8  |63.7  |  41.6 | 60.9 |22.2 | 27.0 | 61.4 |
>
> 4. **Your question on the performance change if no prior knowledge about the categories is given, i.e., the prompt set is disjoint from the GT categories.**
>
>     Please refer to the response and results above (in 2).
>
> 5. **Your question on the K samples' choice and whether the approach works for novel tasks at test time in Section 5.2.**
>
>     Please refer to the response and results above (in 3).
>
> 6. **Your question on FLAIR's performance on COCO-Facet.**
>
>     We attach the results for FLAIR [2] as follows. We also evaluated CLIP-ViT-B/16 in the same scale as this smaller model. We found that FLAIR offers little improvement, suggesting it struggles to focus on the required attribute. This is probably due to its single patch-level alignment, while a concept might align better with several patches jointly.
>
>     *Table 6: Recall@1 of FLAIR and CLIP-ViT-B/16 (baseline) in percentage points on COCO-Facet.*
>     | |Animals | Scenes | Objects | Count of People | Materials | Times | Weathers | Gestures | Avg.|
>     | ---- |:----:|:-----:|:-----:|:-----:|:-----:|:-----:|:-----:|:-----:|:-----:|
>     | CLIP-ViT-B/16 | 94.0 |  44.2 |  57.4  | 1.1  |  1.6  | 3.8 | 3.3 |  9.6 | 35.0|
>     | FLAIR | 82.2 | 52.9 | 54.5  |  1.2  |  3.0  | 6.3  | 4.6 | 6.3 | 33.0|
>
>     *Table 7: Recall@5 of FLAIR and CLIP-ViT-B/16 (baseline) in percentage points on COCO-Facet.*
>     | |Animals | Scenes | Objects | Count of People | Materials | Times | Weathers | Gestures | Avg.|
>     | ---- |:----:|:-----:|:-----:|:-----:|:-----:|:-----:|:-----:|:-----:|:-----:|
>     | CLIP-ViT-B/16 | 99.1| 75.6  |  77.6  | 14.7 |  10.7  | 17.4 |  12.4  | 20.9 |49.8 |
>     | FLAIR | 97.1 | 81.4 |  80.2  |  9.7  |  10.4  |11.8  | 17.9  | 19.5 |50.3 |
>
> If you have further questions or concerns, we would be happy to address them!
>
> [1] Tong et al., Eyes Wide Shut? Exploring the Visual Shortcomings of Multimodal LLMs, CVPR 2024.
>
> [2] Xiao et al., FLAIR: VLM with Fine-grained Language-informed Image Representations, CVPR 2025.

---

> ### Comment · Reviewer_mpkJ · 2025-08-04
>
> I thank the authors for the detailed rebuttal and for making the effort to conduct additional experiments.
>
> The new results are insightful. Prior knowledge seems to play a critical role when it comes to the initially reported performance gains. In the disjoint setting, there are still some gains, but more selectively and not nearly as strong.
>
> I think it would be valuable to discuss these findings as part of a limitation of the proposed method and direction for future work to close the gap between the zero-shot setting and with prior knowledge.
>
> I maintain my positive rating.

---

> > ### Author Response · Authors · 2025-08-05
> >
> > Thank you for your valuable feedback! We will include these tables in the limitation part and describe the retrieval setting more clearly in the main text.

---

### Official Review · Reviewer_tFqQ · 2025-07-03

**Clarity:** 2
**Significance:** 2
**Originality:** 3
**Rating:** 4
**Confidence:** 3

**Summary:**

This paper introduces COCO-FACET, a benchmark for evaluating text-to-image retrievers on attribute-focused queries, revealing limitations in current approaches. The authors demonstrate that both CLIP-like and MLLM-based retrievers struggle with fine-grained visual attributes, particularly with non-dominant elements. As a solution, they propose using promptable image embeddings through MLLM-based universal embedders and present two acceleration strategies to make this approach more practical.

**Questions:**

The underlying principle of the Linear Approximation acceleration method requires clarification. In line 299, the authors propose using Wa for retrieval, which appears to rely solely on image embeddings without a prompt. Then what is the fundamental distinction between W and standard CLIP encoders? More specifically, while W is learned to approximate the nonlinear transformation U(a,p) for a specific prompt p, it's counterintuitive that this linear operation can generalize to highlight specific attributes across the entire image pool for different queries without the prompt being present during retrieval. Further explanation of how this approximation preserves attribute-specific information would strengthen the paper.

**Ethical Concerns:**

["NO or VERY MINOR ethics concerns only"]

**Final Justification:**

After reviewing the authors' rebuttal to other reviewers, I find that their detailed explanation regarding computational overhead addresses most of my concerns. In addition, I would recommend that the authors include more intuitive explanations or visualizations in the appendix to facilitate easier understanding of the proposed Linear Approximation acceleration method for readers. I finally decided to increase my rating to 4.

**Limitations:**

yes

**Quality:**

3

**Strengths And Weaknesses:**

## Strengths

1. Well-constructed benchmark: The COCO-FACET dataset is methodically structured with 9,112 queries across 8 attribute types, providing a valuable resource for the community to evaluate attribute-focused retrieval capabilities.

2. Comprehensive evaluation: The paper presents thorough benchmarking across 12 state-of-the-art retrieval models with varying architectures and scales, offering a clear picture of current limitations in the field.

3. Effective acceleration strategies: The proposed acceleration methods demonstrate meaningful performance improvements.

## Weaknesses

1. Limited technical novelty: The core approach of using MLLMs to obtain multimodal representations has been explored in previous work. The paper's main contribution appears to be the benchmark itself rather than a novel technical solution to the identified problem.

2. Efficiency concerns remain: While acceleration strategies are proposed, the fundamental challenge of computational cost in promptable image embedding approaches persists. The vanilla approach requires N×M forward passes, which remains impractical for large-scale retrieval systems.

3. Conflicting motivations: There is a conflict between the stated goal of improving retrieval for uncommon attributes and the design of the pre-processing embeddings acceleration method, which requires predefined attribute types. This approach only works for a finite set of predetermined attributes, contradicting the motivation to handle diverse, unexpected attribute queries.

---

> ### Author Rebuttal · Authors · 2025-07-31
>
> Thank you for taking the time to review our paper, and we appreciate your comments and suggestions! We address your concerns as follows:
>
> 1. **Your concern on our technical novelty, as using MLLMs to obtain multimodal representations has been explored in previous work.**
>
>     We agree that multimodal representations have been explored previously, but within a limited scope. As stated in lines 95-100 in the Related Work section, previous practices either use multimodal embeddings for encoding given queries that have both image and text (e.g., composed image retrieval like CIRR [1]), or limit the prompts to be domain-specific (e.g., "News, Miscellaneous, and Fashion" in MM-Embed [2]) and hard-coded for a benchmark.
>
>     We are the first to systematically study this method for attribute-focused embedding of target images by using various GPT-written prompts generated through a pipeline, and we reveal its effectiveness, even for some uncommon, challenging attributes. Moreover, the prompts are applied in a zero-shot way, compared with the improvement shown through domain-specific finetuning in MM-Embed [2]. We further proposed two acceleration approaches, showing a wider scope of its application.
>
> 2. **Your concern on the efficiency of our promptable approach.**
>
>     While the vanilla approach requires high computational cost (N×M forward passes), the two acceleration strategies are much more efficient. The pre-processing strategy has the same total per-query cost as the no-prompt approach. For the linear approximation strategy, we only recompute promptable embeddings for K randomly selected images, resulting in a per-query embedding cost of $KF + F$ (lines 305–306), which does not scale with the index size $N$. The cosine similarity search over all $N$ embeddings in the pool still costs $ND$ for exact search (e.g., FAISS IndexFlatIP). Thus, the total per-query cost is $KF + F + ND$, compared to $F + ND$ of the baselines. This $ND$ search term dominates when $N$ is large, which is common in real-world usage.
>
> 3. **Your concern about the conflict between our motivations and the design of the pre-processing embeddings acceleration method.**
>
>     We agree that the pre-processing acceleration method assume a finite attribute vocabulary, which does limit support for entirely novel and highly specific attributes (e.g., "Find me an image with its top-right part in yellow"). This motivated us to design the fallback strategy (on-the-fly linear approximation) for this case.
>
>     Still, the goal of improving retrieval for uncommon attributes holds: we show that MLLM-based retrievers can encode rare attributes more effectively when prompted, and our benchmark enables measuring this. The pre-computing acceleration strategy is a practical compromise to improve scalability for pre-specified, commonly-used attributes, while preserving fallback support for rare or novel ones. Additionally, in lines 286-288 and Table 5, we observe that using the exact ground-truth prompt is often not necessary, as "Count of People" prompt works well on many testcases in "Gesture", showing applicability to unseen but similar attributes.
>
> 4. **Your question about the underlying principle of the Linear Approximation acceleration method.**
>
>     First, we did not argue that this linear operation could recover the full function of the vanilla approach. In lines 308-311, we mentioned that it has limited expressiveness and cannot capture the complex cross-modal interaction happening in the original MLLM-based embedder.
>
>     Second, from the modeling perspective, the linear approximation approach has the same structure with the standard CLIP encoder. Nonetheless, our results show that it can be attribute-focused, analogous to a CLIP model finetuned on attribute-specific data. While our approach only uses K random images from the pool without their labels or actual model training, finetuning CLIP usually requires a large number of images along with captions in the target domain.
>
>     We agree that some visualization like heatmaps showing image focus would help the reader understand the approach better. We will add such visualization in the Appendix.
>
> If you have further questions or concerns, we would be happy to address them as well!
>
> [1] Liu et al., Image Retrieval on Real-life Images with Pre-trained Vision-and-Language Models, ICCV 2021.
>
> [2] Lin et al., MM-Embed: Universal Multimodal Retrieval with Multimodal LLMs, ICLR 2025.

---

> > ### Comment · Reviewer_tFqQ · 2025-08-05
> >
> > I would like to thank the authors for their responses. After reviewing the authors' rebuttal to other reviewers, I find that their detailed explanation regarding computational overhead addresses most of my concerns. In addition, I would recommend that the authors could include more intuitive explanations or visualizations in the appendix to facilitate easier understanding of the proposed Linear Approximation acceleration method for readers. I will increase my rating to 4.

---

> > > ### Author Response · Authors · 2025-08-05
> > >
> > > Thank you for your feedback and for reconsidering your score! We will include the tables on computational overhead analysis, as well as some visualization of the linear approximation approach in the Appendix.

---

### Author Response · Authors · 2025-08-09
**Summary of Reviewer Concerns and Author Responses**

We thank the reviewers for their constructive feedback and are encouraged by the positive recognition of our contributions, including the COCO-FACET benchmark ("a valuable contribution ... encouraging further research" – `roin`), our extensive evaluation of 12 state-of-the-art retrieval models ("a clear picture of current limitations" – `tFqQ`), our promptable approach ("a reasonable step towards solving this task" – `mpkJ`), and the acceleration strategies ("meaningful performance improvements" – `tFqQ`).

We also thank the reviewers for their constructive criticisms, which help us clarify the arguments better. We note that all reviewers expressed that their concerns were largely resolved during the rebuttal. Below, we summarize the major concerns and our responses to them:

### **Computational overhead of the acceleration approaches**

Reviewers `tFqQ` and `Lakj` noted the higher cost of promptable embeddings versus traditional T2I methods, and Reviewer `roin` requested inference times. We clarified that Linear Approximation’s cost does not scale with index size when $K$ is fixed, and that the method remains feasible for large datasets (lines 305-306).

To demonstrate the actual overhead, we measure the inference time on a larger image pool: When $N$ reaches 20M on CPUs, the embedding cost ($K=10$) is no longer the bottleneck. Meanwhile, the memory overhead fits within a single A6000 for $K=100$. Further, we clarify that the time usage and compute are flexible with the available resource: Even with small $K$, the method improves over the baseline, and the memory overhead can be adjusted by varying the batch size. We will include this computational analysis in the revised version.

### **Assumptions of the promptable embedding approach**

Reviewers `mpkJ` and `Lakj` raised concerns about reliance on predefined prompts or prior knowledge about the targeted attributes. First, we agree that the pre-defined prompts limit support for entirely novel attributes, but they still show applicability to similar attributes (lines 286-288). For highly novel attributes at inference, we recommend using the Linear Approximation as a fallback option.

To clarify the scope of our accelerated approaches, we test scenarios removing these assumptions and find smaller but still positive gains over baseline. We will report these results in the Limitations section, and state this assumption about prior knowledge more clearly in the main text.

### **Comparison to concurrent similar methods**

Reviewer `roin` and `mpkJ` asked for clearer distinctions from similar methods (CLOC and FLAIR). These methods impose spatial constraints (bounding boxes or patch-wise alignment) that limit generality for non-region-based attributes. Our promptable approach does not assume the shape of image focus or alignment mode, thus relaxing such constraints.

To better illustrate this point, we evaluate FLAIR on COCO-Facet and find little improvement compared with the CLIP-ViT-B-16 baseline. We will include these results and emphasize the difference in the revised version.

### **Principle of the Linear Approximation acceleration method**

Reviewer `tFqQ` and Reviewer `roin` suggested that the underlying principle of the Linear Approximation requires clarification, and Reviewer `mpkJ` requested more clarity on the pipeline. We will emphasize in the main text that the $K$ samples are uniformly sampled from the corresponding category, and will include visualizations (e.g., heatmaps of the image focus) to illustrate its behavior for readers.

---

### Decision · Program_Chairs · 2025-09-17

**Decision:**

Accept (poster)

**Comment:**

The paper was reviewed by four experts and received lukewarm reviews, albeit leaning towards recommending acceptance. Several of the identified weaknesses were addressed in the authors' response to the satisfaction of the reviewers. These clarifications and the many other suggestions from the reviewers should be considered as the authors prepare a final revision of the paper. However, on the balance the Area Chair believes that the paper (with the promised revisions) meets the bar for acceptance.